# SURE-VQA: Systematic Understanding of Robustness Evaluation in Medical VQA Tasks

**Kim-Celine Kahl[1,2,3*], Selen Erkan[1,2,4*], Jeremias Traub[1,2,4], Carsten T. Lüth[1,2,3], Klaus Maier-Hein[1,2,3,6,7], Lena Maier-Hein[2,3,4,7], Paul F. Jaeger[2,5]**

[1]*German Cancer Research Center (DKFZ) Heidelberg, Division of Medical Image Computing, Germany*
[2]*Helmholtz Imaging, German Cancer Research Center (DKFZ), Heidelberg, Germany*
[3]*Faculty of Mathematics and Computer Science, University of Heidelberg, Germany*
[4]*German Cancer Research Center (DKFZ) Heidelberg, Division of Intelligent Medical Systems, Germany*
[5]*German Cancer Research Center (DKFZ) Heidelberg, Interactive Machine Learning Group, Germany*
[6]*Pattern Analysis and Learning Group, Department of Radiation Oncology, Heidelberg University Hospital, Germany*
[7]*National Center for Tumor Diseases (NCT) Heidelberg, Germany*

*{k.kahl, selen.erkan}@dkfz-heidelberg.de*

**Reviewed on OpenReview:** *https://openreview.net/forum?id=qjNdGpgpV8*

## Abstract

Vision-Language Models (VLMs) have great potential in medical tasks, like Visual Question Answering (VQA), where they could act as interactive assistants for both patients and clinicians. Yet their robustness to distribution shifts on unseen data remains a key concern for safe deployment. Evaluating such robustness requires a controlled experimental setup that allows for systematic insights into the model's behavior. However, we demonstrate that current setups fail to offer sufficiently thorough evaluations. To address this gap, we introduce a novel framework, called *SURE-VQA*, centered around three key requirements to overcome current pitfalls and systematically analyze VLM robustness: 1) Since robustness on synthetic shifts does not necessarily translate to real-world shifts, it should be measured on real-world shifts that are inherent to the VQA data; 2) Traditional token-matching metrics often fail to capture underlying semantics, necessitating the use of large language models (LLMs) for more accurate semantic evaluation; 3) Model performance often lacks interpretability due to missing sanity baselines, thus meaningful baselines should be reported that allow assessing the multimodal impact on the VLM. To demonstrate the relevance of this framework, we conduct a study on the robustness of various Fine-Tuning (FT) methods across three medical datasets with four types of distribution shifts. Our study highlights key insights into robustness: 1) No FT method consistently outperforms others in robustness, and 2) robustness trends are more stable across FT methods than across distribution shifts. Additionally, we find that simple sanity baselines that do not use the image data can perform surprisingly well and confirm LoRA as the best-performing FT method on in-distribution data. Code is provided at `https://github.com/IML-DKFZ/sure-vqa`.

## 1 Introduction

Recent advancements in Vision-Language Models (VLMs) have seen increasing potential for application in the medical domain, with one key area being Visual Question Answering (VQA). In this task, VLMs could assist clinicians or function as medical chatbots for patient inquiries. Several general medical pretrained VLMs, like LLaVA-Med (Li et al. (2023)) and Med-Flamingo (Moor et al. (2023)), have already been developed.

---

*These authors contributed equally to this work

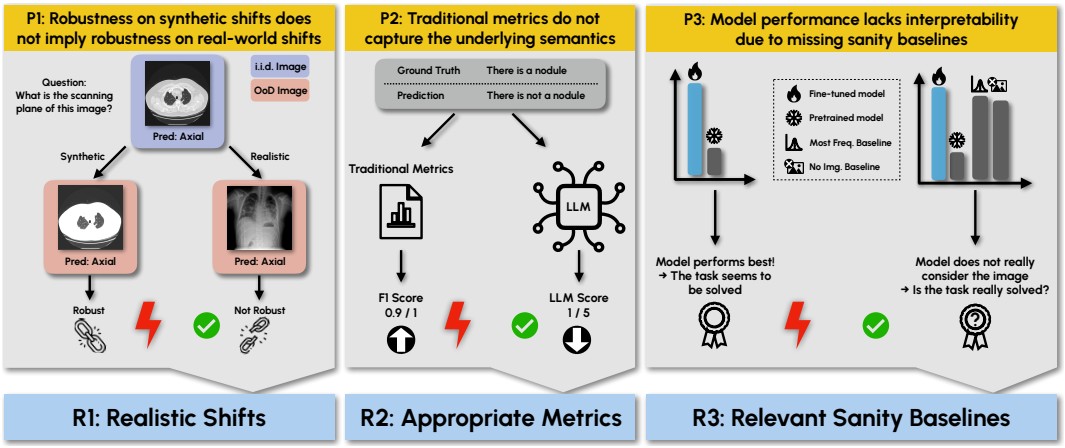

Figure 1: **Pitfalls and Requirements for Systematic Evaluating the Robustness of VLMs in VQA Tasks**. We aim to overcome pitfalls (P1-P3) in the current evaluation of VLM robustness by satisfying the three requirements (R1-R3): We define a diverse set of realistic shifts (R1). We use appropriate metrics for evaluation by using an LLM as evaluator of the VLM output (R2). Finally, we compare the results of the VLM with relevant sanity baselines to see the performance gains over such baselines like considering the text of the question only (R3).

However, a crucial question remains: how robust are these models when faced with variations in data distribution during application? The datasets used for training or fine-tuning may not fully capture the variations in real-world clinical data. As an example, Roberts et al. (2021) highlights how the urgency of the COVID-19 pandemic led to many studies utilizing datasets that insufficiently represent pediatric patients, introducing bias into the analyses. These shifts, whether through unseen disease variations, variations in image acquisition, or different question subjects, may cause performance degradation. Understanding how robust VLMs are to these changes is key to ensuring clinical reliability.

Despite the importance of this research question, existing benchmarks fail to offer an adequate framework to address it effectively. While several benchmarks exist for evaluating the robustness of VLMs under artificial image or text corruptions (Zhang et al. (2024); Chen et al. (2023)), there remains a notable gap in benchmarks that account for more realistic data shifts. We address this gap in the medical domain by utilizing existing medical VQA datasets and setting them up to test VLM robustness against realistic shifts inherent to the VQA data. This is crucial, as prior research has shown that robustness to synthetic shifts does not necessarily translate to robustness under real-world conditions (Taori et al. (2020)). Additionally, many current benchmarks rely on traditional metrics, which use token matching between the ground truth and the model's predictions. We highlight common flaws in these metrics and instead propose the use of large language models (LLMs) as evaluators, validating this approach through a human rater study. Finally, current benchmarks often overlook simple baselines, such as testing a model's ability to answer questions based solely on text. Including such sanity baselines can reveal language priors in the dataset, where questions might be easily answered by their content or by predicting the most common answer seen during training. To overcome these three apparent pitfalls in the current literature, we define three key requirements. Based on these requirements we present a flexible framework, called *SURE-VQA*, that enables meaningful evaluation of VLM robustness in the medical domain.

To showcase the relevance of SURE-VQA, we conduct a study comparing the robustness of various fine-tuning (FT) approaches, similar to Chen et al. (2023), but with a focus on the medical domain. We focus on FT methods, including full FT and parameter-efficient FT (PEFT) because fine-tuning VLMs is crucial and common practice for precision in specialized tasks like medical VQA (Li et al. (2023); Wu et al. (2024); Singhal et al. (2023)). Our study provides valuable insights to the following questions: How does a medical vs. non-medical pretrained VLM perform on the medical VQA task after fine-tuning? How do the FT methods perform in comparison to sanity baselines that, for instance, omit image content? How does the

performance of different FT methods vary across datasets? How to FT methods compare in terms of i.i.d. performance and robustness? Which shift is most severe regarding model robustness?

In summary, our contributions are:

1. Systematically analyze current pitfalls and derive key requirements for meaningful robustness evaluation of VLMs in the medical domain.
2. Provide a flexible open-source framework, named *SURE-VQA*, hosted at: `https://github.com/IML-DKFZ/sure-vqa`.
3. Provide empirical evidence for the derived requirements, including a human rater study that confirms the importance of LLM-based metrics.
4. Show the relevance of SURE-VQA by performing a meaningful comparison of the robustness of FT methods in the medical domain leading to valuable insights for the community.

## 2   Requirements for a systematic evaluation of the robustness of VLMs

We identify several key pitfalls in the current evaluation of VLM robustness, addressed by our framework in Figure 1. In the following section, we detail these pitfalls (P1-P3) and formulate requirements (R1-R3) to overcome them. Finally, we outline our setup to fulfill these requirements.

**P1: Robustness on synthetic shifts does not imply robustness on real-world shifts.** Many existing benchmarks that focus on the robustness of VLMs, such as those in Chen et al. (2023); Shirnin et al. (2024); Qiu et al. (2022), primarily introduce artificial perturbations to the image or text content. A notable exception is Radford et al. (2021), where the robustness of natural distribution shifts is explored in the context of their proposed CLIP model. They find that CLIP is quite effective in being more robust on natural distribution shifts of ImageNet. However, their analysis is limited by CLIP's non-generative tasks and does not consider the generative VQA task. In the medical domain, Nan et al. (2024) conduct a multimodal robustness benchmark. Their study focuses on medical VQA tasks, but it remains limited to testing image corruptions, leaving the crucial question of how realistic distribution shifts impact model performance open. Another study by Jensen & Plank (2022) takes a step toward addressing more realistic data shifts by examining the performance of VLMs on different versions of the VQA dataset, when training them from scratch vs. fine-tuning a pretrained model. However, their focus is primarily on linguistic variations, leaving shifts in image content under-explored. Similarly, even though the benchmark by Zhang et al. (2024) uses artificial image and text corruptions, their content bias might be one step towards more realistic shifts.

However, while the robustness benchmarks in the VLM domain rarely address any realistic shifts, in the unimodal domain there is evidence that models are not robust against natural distribution shifts (Taori et al. (2020); Miller et al. (2020)). This issue has also been proven for fine-tuned, domain-specific models (Yuan et al. (2023)). Furthermore, there is evidence that artificial shifts do not necessarily translate to realistic shifts (Taori et al. (2020)).

→ *R1: Evaluate VLMs under a diverse set of realistic shifts.*

*Implementation in SURE-VQA:* We utilize three datasets from the medical VQA domain, including SLAKE (Liu et al. (2021)), OVQA (Huang et al. (2022)), and MIMIC-CXR-VQA (Bae et al. (2023)). On these datasets, we define several realistic shifts, spanning a range of subtleties, with some having a more pronounced impact, such as modality shifts, while others, like gender shifts, are more subtle. Furthermore, certain shifts primarily affect the image content, such as changes in the body location being imaged, whereas others influence the text input, such as shifts in question types.

**P2: Traditional metrics do not capture the underlying semantics.** Many studies in the VLM field continue to rely on traditional metrics like BLEU and CIDEr (Sung et al. (2022); Chen et al. (2023); Qiu et al. (2022)), or accuracy-based metrics (Li et al. (2023); Jensen & Plank (2022); Qiu et al. (2022)), which are dependent on word- or n-gram matches. We refer to these as "traditional metrics" throughout the paper. Recent research, however, has begun to adopt a more sophisticated approach by employing LLMs to evaluate

the output of VLMs (or other LLMs) (Wang et al. (2024); Ostmeier et al. (2024); Liu et al. (2023b); Chiang & Lee (2023); Fu et al. (2024); Kocmi & Federmann (2023)).

The primary limitation of traditional metrics is their inability to capture the underlying semantics of a sentence. They fail to recognize synonyms or account for negation, often misjudging sentences that differ from the ground truth by a single token, such as "not." Examples of these failures are shown in Appendix A.1, similar to Ostmeier et al. (2024). While we are not the first ones to propose using an LLM as an evaluator, the continued reliance on traditional metrics despite their flaws highlights the need to establish this as an explicit requirement for VLM evaluation.

$\rightarrow$ *R2: Evaluate VLMs with appropriate metrics that capture the underlying semantics of the output.*

*Implementation in SURE-VQA:* We employ the Gemma model (Team Gemma et al. (2024)) as an evaluator, utilizing three distinct prompts tailored for different question types: open-ended, closed-ended binary, and closed-ended multilabel. Details on the question types can be found in section 3.1. To optimize computational efficiency and reduce potential errors from the LLM evaluator, we implement a hybrid metric. Specifically, when the answer exactly matches the ground truth, we assign the highest score without invoking the LLM, saving computational resources and minimizing the risk of evaluation failures. Additionally, we assess the feasibility of this evaluation by conducting a human rater study, where we empirically validate its performance in comparison to traditional metrics and other LLMs.

**P3: Model performance lacks interpretability due to missing sanity baselines.** Currently, the performance of VLMs is typically reported either in isolation or in comparison to other VLMs. This is evident in papers that introduce new VLMs (Li et al. (2023); Moor et al. (2023)), and in benchmark studies (Chen et al. (2023); Zhang et al. (2024); Qiu et al. (2022); Nan et al. (2024)). An exception is the work of Liu et al. (2023a), where the performance of a language-only GPT-4 is evaluated. However, their focus is to improve the model by ensembling with the LLM rather than to highlight a potential issue in current VQA datasets that many questions can be answered using only text information. Further, they do not provide a *no image* sanity baseline of their own model, leaving its multimodal usage unexplored. Another study by Parcalabescu & Frank (2023) contextualizes the multimodal use of VLMs by employing Shapley values to assess the contribution of each modality to the output. However, this method is computationally more complex during inference, since multiple forward passes are needed to estimate the Shapley values.

The problem with many VQA datasets is that they tend to contain hidden patterns, allowing models to exploit shortcuts (Geirhos et al. (2020)) rather than using all available information, including the image content (Kafle & Kanan (2017); Chen et al. (2024a); Kervadec et al. (2021); Goyal et al. (2017); Dancette et al. (2021)). This means that high performance on these datasets does not guarantee that the model is actually utilizing the visual input to answer the question; instead, it might be exploiting patterns in the questions (Kafle & Kanan (2017); Kervadec et al. (2021)). As demonstrated by Chen et al. (2024a), in many cases, the visual content in VQA datasets is unnecessary, and models can achieve high performance simply by relying on the text. This indicates that the models are leveraging hidden biases in the question-answer pairs rather than solving the task as intended.

$\rightarrow$ *R3: Provide relevant sanity baselines to contextualize the benefits of VLM fine-tuning and multimodal information usage.*

*Implementation in SURE-VQA:* We propose that using relevant sanity baselines to reveal such dataset biases can be beneficial to explore how the models solve the given task and how the datasets are structured. Thereby, we put the performance of the fine-tuned model into context by comparing it to baselines in two aspects: (1) Not using the image information: Here, we *a)* choose the most frequent answer to the question in the training set and answer the same question in the test set with this (*most frequent* baseline) and *b)* train the model without using any image information, which should lead to learning shortcuts based on the language (*no image* baseline). (2) Not fine-tuning the model: Use the plain VLM in a zero-shot setting without any fine-tuning and report the performance on the test set. This serves at the same time as a baseline to see if or how much the shifts are inherently different between i.i.d. and OoD when not fine-tuning.

# 3 Empirical Study

To provide evidence of the relevance of our framework, we perform experiments to (1) empirically validate the necessity of requirements R1–R3, and (2) show that our framework allows valuable insights into robustness by conducting an extensive empirical study on the robustness of FT methods. In this section, we first present the datasets and experimental setup used in our studies, followed by a comprehensive analysis and discussion of the empirical findings.

## 3.1 Utilized Datasets

An overview of the utilized datasets is provided in Figure 2, with further details regarding the datasets, preprocessing steps, and split sizes available in Appendix C.1. In total, we use three different medical VQA datasets, each with closed-ended and open-ended questions. Closed-ended questions are questions where a fixed set of options is given as part of the question (e.g. "yes"/"no", "CT"/"MRI"), while open-ended questions are not constrained to certain options in the answer. Each dataset incorporates a variety of realistic shifts to meet R1. The taxonomy for shift categories is thereby partially taken from Castro et al. (2020), also used in related work such as Bungert et al. (2023); Roschewitz et al. (2023); Choi et al. (2023).

**SLAKE**  We use the SLAKE dataset (Liu et al. (2021)) with two different shifts: 1) *Modality shift*: Representing an acquisition shift (Castro et al. (2020)), we train the model exclusively on CT and MRI images (2D slices) and then test it on X-ray images. 2) *Question type shift*: During training, the model is exposed to questions about image content such as shape and color but excludes questions related to the size of organs. These size-related questions are introduced in the OoD test set.

**OVQA**  The OVQA dataset (Huang et al. (2022)) is used with two shifts: 1) *Body part shift*: Representing a manifestation shift (Castro et al. (2020)), we train the model on images of the hand, chest, and head, and test it on images of the leg. 2) *Question type shift*: In the training set, the model is exposed to questions about various image contents like abnormalities and conditions, but questions related to the organ system are reserved for the OoD test set.

**MIMIC-CXR-VQA**  We use the MIMIC-CXR-VQA (Bae et al. (2023)) dataset with three different shifts, all representing population shifts (Castro et al. (2020)): 1) *Gender shift*: The model is trained on male patients and tested on female patients. 2) *Population shift*: Training is conducted using data from white patients, with testing on patients from other ethnicities. 3) *Age shift*: The model is trained on patients over the age of 60 and tested on patients under 40. A gap is intentionally introduced between the i.i.d. and OoD groups to make the shift more explicit.

## 3.2 Study Design

We utilize above mentioned VQA datasets, splitting them so that the training and testing distributions differ. We employ two different base models: LLaVA-Med 1.5, a state-of-the-art medical VLM (Li et al. (2023)), and its corresponding non-medical base model, LLaVA 1.6 (Liu et al. (2023a)). For fine-tuning, we use the following methods: full FT, prompt tuning (Lester et al. (2021)), LoRA (Hu et al. (2021)), and $(IA)^3$ (Liu et al. (2022)). Hyperparameters for the PEFT methods are selected based on the full training set and corresponding validation set for each dataset. Details regarding the hyperparameter search can be found in Appendix C.2. To measure robustness, we split the data into i.i.d. training and i.i.d. and OoD test sets, as outlined in section 3.1, thereby fulfilling R1. We then measure the performance of the VLMs using the language model Gemma, fulfilling R2. The selection of Gemma as an evaluator is further justified in section 3.3.2. For robustness measurement, we calculate the *relative robustness (RR)* (Chen et al. (2023)), defined as $RR = 1 - \Delta P/P_I$, where $\Delta P = (P_I - P_O)$, and $P_I$ is the i.i.d test performance and $P_O$ is the OoD test performance. For a better interpretation of the results, we compare them against relevant sanity baselines as described in R3.

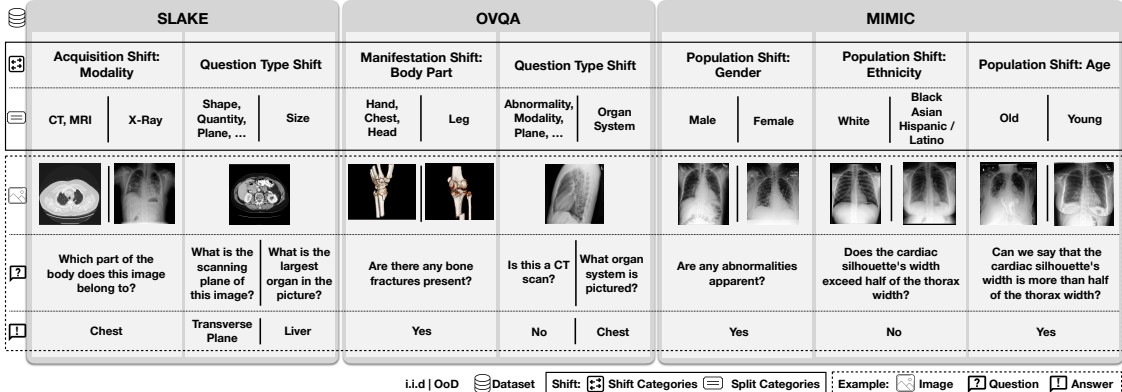

Figure 2: **Datasets and Shifts Used in the Study**. We use three datasets with four different types of shifts, resulting in seven different settings for robustness analysis. Shifts that are mainly focused on changes in the image content are shown by a change of images between i.i.d. and OoD and shifts that focus on the question content are shown by a change of the question and answer between i.i.d. and OoD shifts. The taxonomy for the shift category is partially taken from Castro et al. (2020).

## 3.3 Empirical Confirmation of R1-R3

### 3.3.1 R1: Comparison between Realistic Shifts and Corruption Shifts

**Study Design** To compare the effect of realistic shifts (R1) against artificial shifts (corruption shifts), we perform a robustness study on the SLAKE dataset, using a simplified setup that involves only the LLaVA-Med model fine-tuned on SLAKE with $(\texttt{IA})^3$. This study aims to support the claims outlined in (R1) within the context of our dataset setup. As illustrated in Figure 20, the model is first fine-tuned on an i.i.d. training set. Then, we test it on the i.i.d. test set and two different OoD test sets—one reflecting realistic shifts, and the other containing artificially corrupted images. For realistic shifts, we use the original i.i.d. and OoD splits defined in section 3.1. For artificial (corruption) shifts, the OoD test set is generated by applying image corruptions like blur, brightness, and noise at varying severity levels to the i.i.d. test images. Each corruption is applied with a probability of 0.5, and at least one corruption is applied per image.

**Results** As seen in Figure 3, the realistic modality and question type shifts exhibit lower relative robustness compared to artificial shifts (medium strength), meaning artificial shifts fail to accurately capture the challenges presented by realistic shifts. This finding highlights the importance of evaluating robustness under realistic conditions, as adopted in our study. While corruption-based shifts may offer a valuable complementary perspective, these results emphasize that they should not be considered the only source for evaluating robustness. Further details and experiments on other corruption levels are provided in Appendix D.

### 3.3.2 R2: Human Rater Study

**Study Design** To ensure that the scores assigned by the LLM align with human judgment, we conduct a human rater study. We use this to compare various LLMs for rating and select the one that aligns best with human judgment. For each dataset, we randomly selected 100 open-ended questions where the prediction did not exactly match the ground truth, as exact matches would automatically score highest by the hybrid metric (R2). Five human raters evaluate the questions, and we calculate the correlation between humans and the LLM using Kendall's Tau (Kendall (1945)). Additionally, we report the correlation between humans and traditional metrics, and the inter-rater variability.

**Results** Among the LLMs evaluated, Gemma demonstrates the highest correlation with human ratings, as detailed in Figure 10 (Appendix B.1). Thus, Gemma is selected as the evaluator, and we present the results of the human rater study using Gemma in the following section. The correlations are shown in

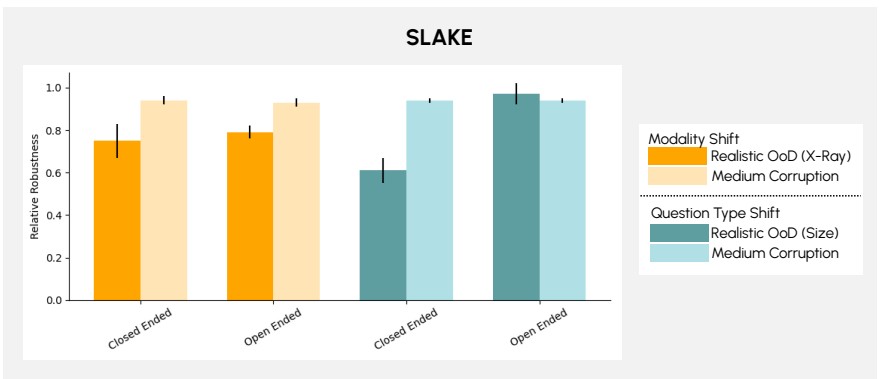

Figure 3: **Comparison between Realistic shifts and Corruption shifts on SLAKE Dataset Fine-tuned on** $(\text{IA})^3$. For realistic shifts, we use the same i.i.d. and OoD training and test sets as described in Section 3.1. For corruption shifts, the i.i.d. training and test sets remain the same as in the realistic setup. The only difference is the OoD test set, which is created by adding one or more corruptions such as blur, brightness changes, or noise to the i.i.d. test images at varying levels of severity.

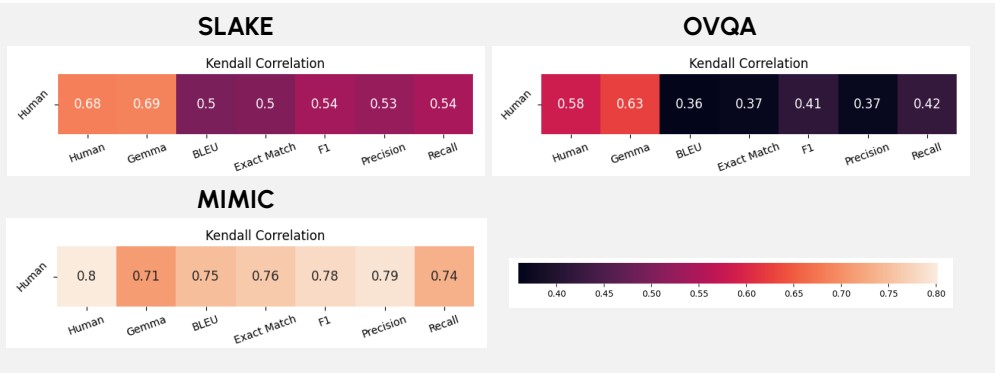

Figure 4: **Results of the Human Rater Study**. Human interrater correlation is calculated between five raters. We use Kendall's Tau Kendall (1945) as correlation measure.

Figure 4, with detailed results available in Appendix B.3. Gemma demonstrates the highest correlation with human ratings on the SLAKE and OVQA datasets, with differences between Gemma and the best traditional metrics being 0.15 and 0.21, respectively. Interestingly, on these datasets, the human interrater correlation is not the highest compared to correlation with other metrics. This suggests that the metrics fall between the evaluations of the human raters, resulting in a stronger correlation with the average human rating than between the individual raters themselves. On the MIMIC dataset, traditional metrics perform slightly better, with the performance gap between Gemma and the traditional metrics ranging from 0.03 to 0.08. This variation can be attributed to several factors. In the OVQA dataset, for example, the structure of the questions and answers makes traditional metrics more prone to failure. Tokens like "fracture," "left"/"right", or specific bone names often match between the ground truth and predictions, despite significant differences in other tokens (see Figure 11a for an example). On the other hand, MIMIC answers are highly structured, particularly for open-ended questions, where a fixed set of classes is listed in a comma-separated format. In these cases, traditional metrics perform well because token matching tends to be more accurate especially when the classes have distinct tokens.

Despite these limitations, Gemma's performance on the MIMIC dataset remains strong, and its correlation with human ratings is even higher than on the other two datasets. Moreover, the failures of Gemma on MIMIC are not complete failures, as shown in Figure 11b. In summary, although there are instances where traditional metrics show higher correlations with human scores, Gemma proves to be generally more robust and less prone to complete failures. Its correlation remains consistently high across all datasets, confirming it as a reliable metric for evaluation in VQA tasks. However, even though Gemma shows high correlation with human judgments, it is important to note that LLMs may carry internal biases. These biases might come from underrepresentation of detailed medical knowledge, recent medical advancements, and diverse socio-demographic groups in LLMs training procedure and data. As a result, such factors can influence the model's scoring behavior in evaluation tasks. That's why it is crucial to conduct human-rater studies and choose an LLM that aligns closely with human judgment to minimize these effects especially when applied to a new use case.

### 3.3.3 R3: Performance of Sanity Baselines

**Study Design**   To evaluate the performance of sanity baselines, we compare a VLM not using the image information (*no image* sanity baseline) with a VLM using the image information. Specifically, we investigate how often the model using the images actually outperforms the model that is not using the images. This allows us to evaluate the model's susceptibility to shortcut learning, where it may rely predominantly on language cues while neglecting visual information. While this section analyses the *no image* baseline in detail, the general confirmation of the other sanity baselines is part of our large-scale study in section 3.4.

**Results**   As seen in Figure 5, overall, the sanity baseline of not using the image information performs surprisingly well. Ideally, in the absence of shortcut learning, the model utilizing image information should almost consistently outperform its image-agnostic counterpart or at least tie on the individual questions. However, this is not the case on any of the datasets. While the win-to-loss ratio is overall positive for the model using the image information, the margin is narrower than anticipated. Most notably, on the MIMIC dataset, the model using the image information is even outperformed on up to 25% of the questions. While on the SLAKE dataset, the win-to-loss ratio is highest for the model using the image information, it is the worst on the MIMIC dataset, and on the OVQA dataset, the models tie the most. Further results on the other sanity baselines as well as potential reasons for performance differences between datasets are provided in section 3.4.

### 3.4 Empirical Study on the Robustness of Fine-Tuning Methods

To show the relevance of SURE-VQA we perform an empirical study comparing the robustness of various FT methods under realistic shifts in medical VQA. This is important since practitioners should be informed about the differences between the methods not only in terms of their performance but also how robust they are when selecting a method. The results of the FT robustness study can be seen in Figure 6, Figure 7 and

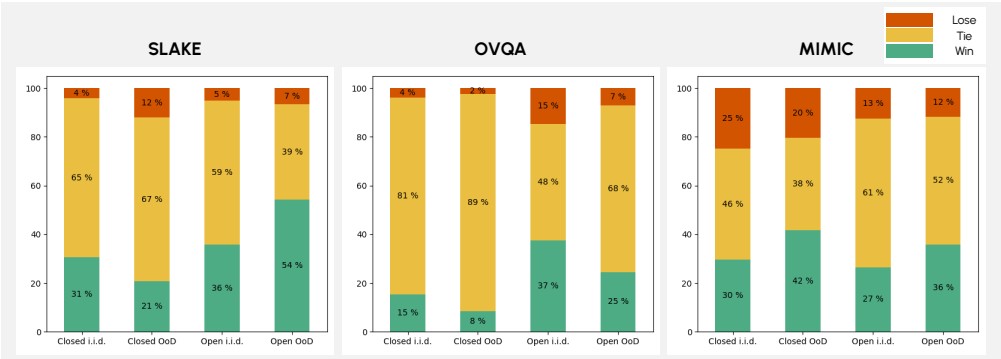

Figure 5: **Comparison between the Model using the Images and the Model not using the Images**. For each PEFT method, the answer scores of one model using the images are compared to the corresponding answers of the model not using the images. Lose = Model using the images loses, Tie = Both answers are equally good, Win = Model using the images wins. Closed i.i.d. / OoD = Closed-Ended questions, Open i.i.d. / OoD= Open-Ended questions. Figure layout inspired by Jeong et al. (2024)

Figure 8. Detailed tables with the results are shown in Appendix C.5. Further, an ablation on the impact of multimodal shifts is shown in Appendix F.

**Comparison between Medical and Non-Medical Base Models.** Figure 6 shows the comparison between the medical (LLaVA-Med 1.5) and the non-medical base model (LLaVA 1.6), aggregated over all FT methods. The results indicate that medical and non-medical models often tie in their answers, with ties ranging from 64% to 90%. On SLAKE and OVQA, the medical model generally achieves a slightly higher number of wins. On the MIMIC dataset, the medical model is slightly outperformed on closed-ended questions but shows marginally better performance on open-ended ones. However, the differences in wins and losses between the two models are mostly minor. This suggests that there is no clear winner between medical and non-medical base models when fine-tuning. Similar results have been shown by Jeong et al. (2024) in zero- and few-shot settings, however, in their experiments, the medical model performed slightly worse. To make the discussion about the following results easier to parse, we detail the results of the medical base model in the main paper and show detailed results on the non-medical base model in Appendix C.4.

**Comparison Between Datasets Considering the Sanity Baselines** Generally, the PEFT models outperform the *no fine-tuned* models, the *most frequent* baseline, and their respective *no image* baselines on the i.i.d. datasets. This is in contrast to full FT, which often does not outperform the *most frequent* baseline. Notably, while the *no image* baseline and the *most frequent* baseline both mostly perform better than random classifier, this effect is particularly noticeable for the closed-ended questions on OVQA, where the *no image* baseline achieves on average 72%, and the *most frequent* baseline 73.6%. Consequently, the gap between the *no image* baselines and the PEFT models is smaller for OVQA (18%) compared to SLAKE (33%). This indicates that the model can often rely on learned question-answer correlations without needing the images, likely because the OVQA dataset has the least unique questions (see Table 4, Appendix C.1.4). For the MIMIC dataset, the gap between the *no image* baseline and the PEFT methods is even smaller than for OVQA, with only an improvement of 11% for the PEFT methods over the baselines. However, as Table 4 suggests, MIMIC does not have many repetitive questions. Thus, the small gap between PEFT models and the *no image* baseline might mean that image encodings contribute minimally to the VQA task. This is likely because the MIMIC dataset focuses on detailed chest X-ray images, and the image encoder may lack this fine-grained expertise. Regarding the no-finetuned model baseline, PEFT models outperform the no-finetuned model on the closed-ended questions by the largest margin on OVQA, with a 43% improvement on average. However, this trend differs on the open-ended questions, where the difference between the fine-tuned models and the no-finetuned model seems larger on SLAKE and MIMIC. To summarize, the models overall seem to learn most on the SLAKE dataset, which also shows the highest i.i.d. performance on the

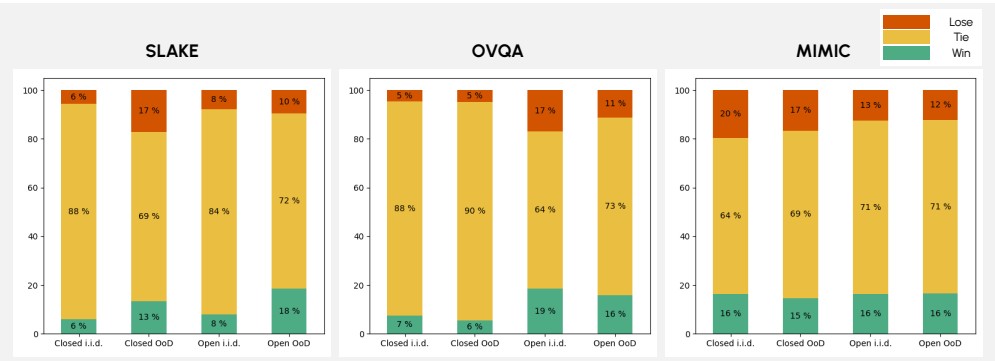

Figure 6: **Comparison between the Medical and the Non-Medical Base Model**. For each FT method, the answer scores of one medical experiment are compared to the corresponding answers of the non-medical experiment. Lose = Medical model loses, Tie = Both answers are equally good, Win = Medical model wins. Closed i.i.d. / OoD = Closed-Ended questions, Open i.i.d. / OoD= Open-Ended questions. Figure layout inspired by Jeong et al. (2024).

PEFT methods with an average accuracy of 86.2% and a Gemma score of 4.3. In contrast, the closed-ended performance on the MIMIC dataset seems generally insufficient for practical use (61%).

**Comparison Between FT Methods.**  When comparing full FT with PEFT methods, we see that full FT neither outperforms PEFT methods in terms of performance nor robustness, aligning with the observations of Dutt et al. (2024) that PEFT methods are particularly well-suited for medical and low-data scenarios. From the PEFT methods, on the i.i.d. set, LoRA is consistently the best PEFT method, achieving an average accuracy of 80.6% on closed-ended and a Gemma score of 3.6 on open-ended questions across different datasets and shifts. However, the performance differences between LoRA and other PEFT methods are quite small, especially considering that these other methods require fewer parameters to train. Overall, robustness within methods is more homogeneous, while there is more variance across dataset shifts. This suggests that the type of shift has a greater impact on robustness than the choice of the fine-tuning method as visualized in Figure 8c). Only on the OVQA dataset, the inter-method variability is higher than the inter-shift variability, but this is due to the failure in robustness of full FT. Only comparing the PEFT methods, also for OVQA the inter-shift variability is lower, shown in Appendix C.5, Figure 18. Besides that, none of the fine-tuning methods consistently outperforms the others in terms of robustness as seen in Figure 8b), where the rank of each method is depicted with respect to their RR. As one exceptional outlier of the PEFT methods, on closed-ended questions in SLAKE, LoRA demonstrates significantly lower robustness compared to other methods, with an RR of 48%, compared to 65% for prompt tuning and 68% for $(\texttt{IA})^3$.

**Comparison Between Shifts.**  The robustness trends vary between datasets, as shown in Figure 8a). On SLAKE, models demonstrate greater robustness on open-ended questions compared to closed-ended ones, while the opposite is true for OVQA. This behavior results from the related question patterns between OoD questions and the training data. Although the OoD questions are not part of the training set, similar training questions offer enough context for generalization, seen in SLAKE for open-ended questions and in OVQA for closed-ended ones. Similar analysis on a conducted ablation study is provided in Appendix E. This pattern is especially clear in question type shifts, where RR on SLAKE is 54% for closed-ended questions and increases to 98% for open-ended. In contrast, on OVQA, RR is 94% for closed-ended questions and drops to 63% for open-ended ones. This means that the question type shift seems most severe if the model performance drops. The population shifts on MIMIC did not affect the models' robustness, i.e. the models seem robust against such shifts since they show over 100% RR. However, since the performance on the MIMIC dataset is generally insufficient, it is questionable if this observation would hold on higher i.i.d. performance. Future experiments could investigate whether the low performance or the kind of shift is causing this behavior.

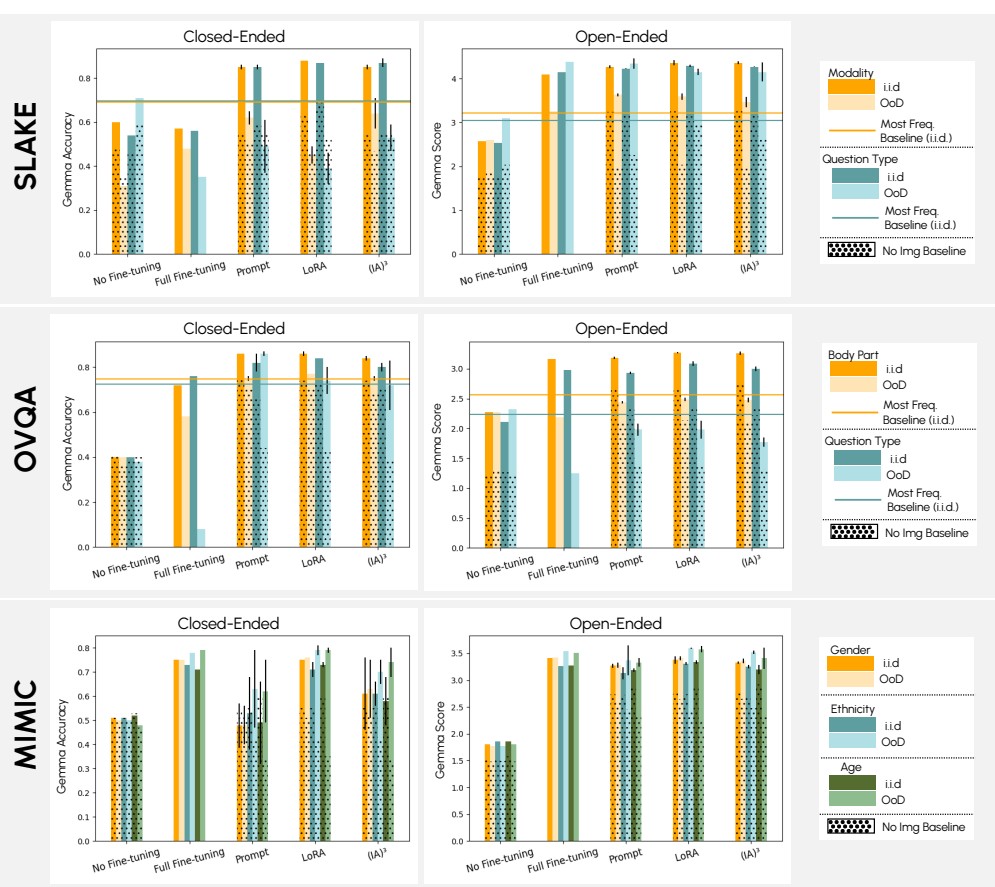

Figure 7: **Results of the FT Robustness Study on the i.i.d. and OoD Test Set**. Reported results show the mean over three seeds (exception: no FT, full FT) with the standard deviation for the non-baselines. Gemma Accuracy refers to the accuracy being rated by Gemma. For MIMIC, the *most frequent* sanity baseline can not be calculated as too few questions match the training set.

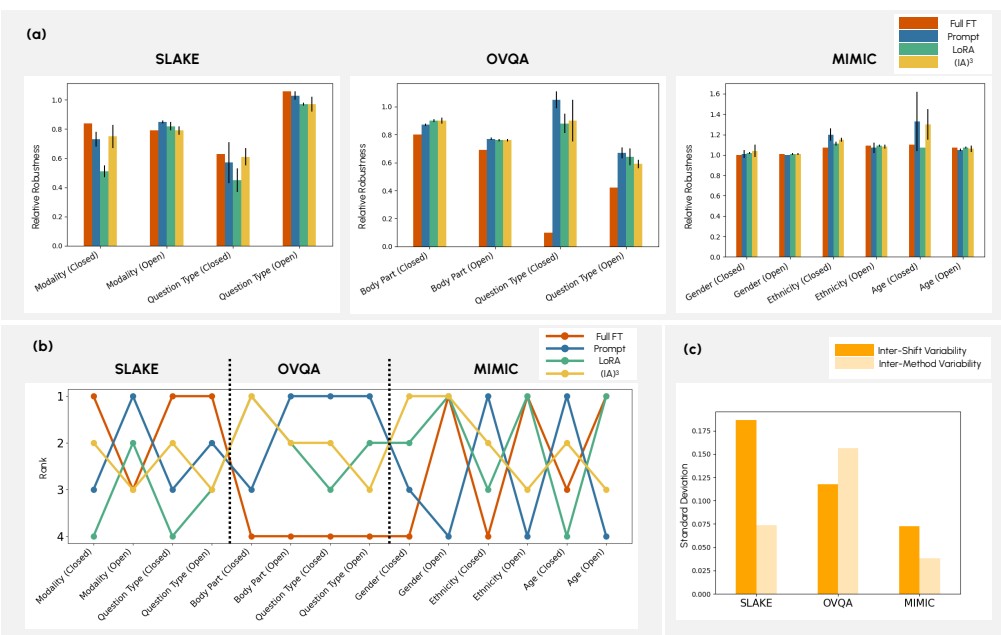

Figure 8: **Results of the FT Robustness Study Focusing on the Relative Robustness (RR)**. Since the study focuses on comparing the FT methods and our definition of OoD holds for these methods, the *no FT* baseline is excluded. (a) RR on the three datasets for the FT methods. Results show the mean and standard deviation over three seeds (exception: full FT). (b) RR Ranking of all FT methods. (c) Standard deviation between shifts vs. standard deviation between FT methods.

## 4    Conclusion and Take-aways

We present a framework that allows testing the robustness of VLMs in medical VQA tasks. Thereby, we especially focus on three key requirements for a meaningful evaluation of robustness.

**Empirical Confirmation of R1-R3.**    While in section 2 we derived R1-R3 from flaws in the current literature, our study provides empirical evidence for their importance. **R1:** In a study (section 3.3.1 and Appendix D), we show that corruption shifts do not necessarily translate to real-world shifts, thereby justifying our claim to also work with more real-world shifts. **R2:** We present several critical failures of traditional token-matching metrics and prove the applicability of our LLM evaluation setup by a human rater study in section 3.3.2. **R3:** We show that some sanity baselines that do not use the image information already perform surprisingly well (section 3.3.3 and section 3.4). This highlights two aspects: 1) As stated in R3, reporting such sanity baselines is crucial for understanding the true multi-modal performance of a VLM, beyond its language-only capabilities. 2) This observation further suggests that we need more elaborate datasets and tasks in medical VQA that minimize the potential for shortcut learning based solely on the language content. Achieving this may involve incorporating greater linguistic variability in questions, as seen in Bae et al. (2023), where questions were rephrased using GPT-4, and ensuring a broad range of semantic differences in the questions. Progress can be tracked as a low performance of the no-image sanity baseline.

**Generalizability of the Framework.**    SURE-VQA serves as a starting point for a comprehensive evaluation of robustness and it can be flexibly extended to new datasets, methods, and domains. Additionally, SURE-VQA can support method development aimed at enhancing the robustness of VLMs. In our study, we define OoD as a data shift w.r.t the FT data. However, our framework also allows to compare VLMs in a zero-shot setting without any FT, when simply redefining OoD as a data shift w.r.t the pre-training data. However, such a definition becomes increasingly challenging to validate with foundation models, since the

exact training data used is often not known. Notably, R2, and R3 go beyond robustness analysis and should be integral to any well-designed VLM study.

**Main Insights from the FT Robustness Study.** Our exemplary study which compares the robustness of various FT methods reveals several key insights. **With regard to robustness**, we find that no single FT method consistently outperforms the others in terms of robustness. Furthermore, robustness trends appear to be more consistent within FT methods than across different dataset shifts, indicating that the type of shift has a greater impact on robustness than the choice of FT method. This suggests that robustness alone is not a decisive factor when choosing a FT method. Additionally, the models are generally robust to population shifts. However, further investigation is needed to determine whether this is due to low i.i.d. performance or because of the nature of the shift. **Besides these insights into robustness**, our comparison of medical and non-medical base models shows that both perform similarly when fine-tuned. Finally, in line with findings of Dutt et al. (2024), we find that PEFT methods are more efficient than full FT. Specifically, we confirm LoRA as the best-performing FT method on the i.i.d. dataset.

**Future Work.** As mentioned above, SURE-VQA is a starting point serving as a foundational framework and is not yet intended as a clinically deployable benchmark for medical VQA systems. Instead, it functions as a research tool to inform and guide future developments in the field. Future work can include the investigation of more datasets, shifts, methods, and models. Particularly, the exemplary study in this paper mainly focuses on radiological VQA datasets. Other medical image types like histopathology, surgical, or dermatology images could be investigated in future work as they may have different characteristics and thus may reveal other findings. Further, while this paper focuses on simple, computational efficient baselines that omit image input, future work could use Shapley values to gain additional insights into the role of images in the VQA task (Parcalabescu & Frank (2023)). Another key research direction is the development of additional VQA datasets, particularly in the medical domain. This would cover two needs: 1) The need for more diverse question types, mitigating shortcut learning during fine-tuning based on the language, and improving the generalization on unseen data and 2) improve the clinical relevance. While current datasets cover various question types, they might offer limited value to clinicians. For example, clinicians might find questions on prognosis and tumor spread more beneficial than scanning modalities. Therefore, collaborating with clinicians can help incorporate more clinically meaningful questions. Finally, the underperformance of two state-of-the-art VLMs, LLaVA and LLaVA-Med on the MIMIC-CXR-VQA dataset underscores the critical need for advancing the development of medical VLMs, particularly given that LLaVA-Med is specifically designed for the medical domain. Even though recent work by Chen et al. (2024b) has focused on developing foundation models specifically for chest X-ray data, there is still a need to develop more robust models capable of handling multiple modalities across a wider range of clinical scenarios.

## Acknowledgements

This work was partly funded by Helmholtz Imaging (HI), a platform of the Helmholtz Incubator on Information and Data Science. We thank Dasha Trofimova for her help in the human rater study as well as insightful discussions and feedback.

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

# A    Evaluation Details

## A.1    Failures of traditional metrics

Examples of failures of traditional token-matching metrics are shown in Figure 9.

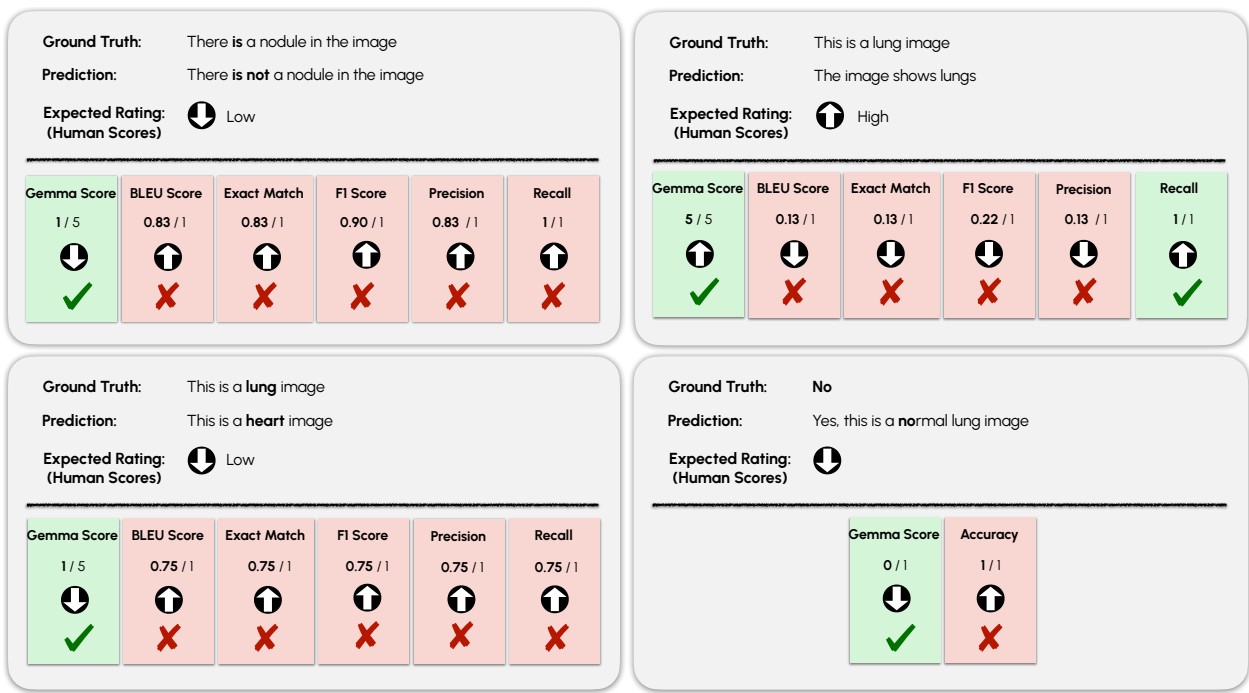

Figure 9: Failures of Traditional Metrics

## A.2  Prompts for Evaluation

Listing 1: Gemma Prompt for Evaluating Open-Ended Questions

```
You are a helpful evaluator to evaluate answers to questions about biomedical
    images.
Score the following answer to a question about an image with respect to the
    ground truth answer with one to five stars.
Where the stars have the following meaning:
 1. One Star: "Incorrect"
   - The answer does not match the ground truth and contains   significant
       inaccuracies.
   - Demonstrates a clear misunderstanding or misinterpretation of the question.
 2. Two Stars: "Partially Correct"
   - The answer has some elements that match the ground truth, but there are
       notable discrepancies.
   - Shows partial understanding but lacks overall accuracy in addressing the
       question.
 3. Three Stars: "Mostly Correct"
   - The answer aligns with the ground truth to a reasonable extent, but there
       are some inaccuracies or gaps.
   - Demonstrates a moderate understanding but may lack
 4. Four Stars: "Correct with Minor Deviations"
   - The answer is largely accurate and corresponds closely to the ground truth.
   - Minor deviations or omissions are present but do not significantly impact
       the overall correctness.
 5. Five Stars: "Perfect Match"
   - The answer exactly matches the ground truth with no discrepancies.
   - Demonstrates a precise and complete understanding of the question,
       providing a flawless response.
Here are some instructions on the input and output format:
 - The input will be passed as json format with the following fields that are
     important:
   - "question": the question about the image
   - "gt": the ground truth answer to the question
   - "pred": the predicted answer to the question
 - The output should be in json format and look the following:
   { score: <xxx>}
   where <xxx> is the number of stars you give to the answer. Do not add
       anything else to the answer.
Input:
```

Listing 2: Gemma Prompt for Evaluating Closed-Ended Questions

```
You are a helpful evaluator to evaluate answers to questions about biomedical
    images.
Score the following answer to a question about an image with respect to the
    ground truth answer with zero or one star.
The questions are all close-ended, so the answer is either correct or incorrect,
     but minor variations in phrasing or acceptable synonyms should still count
    as correct if the core meaning remains unchanged.
Evaluate whether the prediction (pred) accurately matches the ground truth (gt)
    based on meaning and relevance.
The stars have the following meaning:
 0. Zero Star: "Incorrect"
   - The predicted answer is incorrect.
   - The main entity or concept from the ground truth is not correctly
       identified in the prediction
 1. One Star: "Correct"
   - The predicted answer is correct.
   - The main entity or concept from the ground truth is correctly identified in
       the prediction
   - the prediction provides the same information or identifies the same entity/
       concept as ground truth even if it includes additional, irrelevant details
       .
 - Ensure that unrelated phrases or extra descriptions in the prediction do not
     distract from the evaluation of its correctness.
 - Here are some instructions on the input and output format:
```

```
  - The input will be passed as json format with the following fields that are
     important:
   - "question": the question about the image
   - "gt": the ground truth answer to the question
   - "pred": the predicted answer to the question
 - The output should be in json format and look the following:
   { score: <xxx>}
   where <xxx> is the number of stars you give to the answer. Do not add
     anything else to the answer.
Input:
```

Listing 3: Gemma Prompt for Evaluating Closed-Ended Multilabel Questions

```
You are a helpful evaluator to evaluate answers to questions about biomedical
    images.
Score the following answer to a question about an image with respect to the
    ground truth answer with 0, 0.5 or 1 star.
Each question asks for two options in the image and the answer can either be one
    of the options, both of the options or none.
The questions are all close-ended, so the answer is either correct or incorrect,
    but minor variations in phrasing or acceptable synonyms should still count
    as correct if the core meaning remains unchanged.
Evaluate whether the prediction (pred) accurately matches the ground truth (gt)
    based on meaning and relevance.
The stars for rating have the following meaning:
 0 Star: "Incorrect"
  - The predicted answer is incorrect.
  - The main entity or concept from the ground truth is not correctly identified
       in the prediction.
   - This is the case if
     - Option A is the ground truth answer, but the prediction is Option B
     - Option B is the ground truth answer, but the prediction is Option A
     - The ground truth answer is "both", but the prediction is "none"
     - The ground truth answer is "none", but the prediction is "both"
 0.5 Star: "Partially Correct"
  - The predicted answer is partially correct.
  - The main entity or concept from the ground truth is partially correctly
      identified in the prediction.
   - This is the case if
     - Option A/B is the ground truth answer, but the prediction is "both"
     - Option A/B is the ground truth answer, but the prediction is "none"
     - The ground truth is "both", but the prediction is option A/B
     - The ground truth in "none", but the prediction is option A/B
 1 Star: "Correct"
   - The predicted answer is correct.
   - The main entity or concept from the ground truth is correctly identified in
        the prediction
   - The prediction provides the same information or identifies the same entity/
      concept as ground truth even if it includes additional, irrelevant details
      .
   - This is the case if
     - Option A is the ground truth answer and the prediction is Option A
     - Option B is the ground truth answer and the prediction is Option B
     - The ground truth is "both" and the prediction is "both"
     - The ground truth is "none" and the prediction is "none"

Especially for the "none" Cases:
    When the ground truth is "none":
        If the prediction is "none", the score should be 1 star.
        If the prediction is "both", the score should be 0 stars.
        If the prediction is Option A or B, the score should be 0.5 stars.
    When the prediction is "none":
        If the ground truth is "none", the score should be 1 star.
        If the ground truth is "both", the score should be 0 stars.
        If the ground truth is Option A or B, the score should be 0.5 stars.

Especially for the "both" Cases:
    When the ground truth is "both":
```

```
            If the prediction is "both", the score should be 1 star.
            If the prediction is "none", the score should be 0 stars.
            If the prediction is Option A or B, the score should be 0.5 stars.
       When the prediction is "both":
            If the ground truth is "both", the score should be 1 star.
            If the ground truth is "none", the score should be 0 stars.
            If the ground truth is Option A or B, the score should be 0.5 stars.

- Ensure that unrelated phrases or extra descriptions in the prediction do not
     distract from the evaluation of its correctness.
Here are some instructions on the input and output format:
 - The input will be passed as json format with the following fields that are
     important:
    - "question": the question about the image
    - "gt": the ground truth answer to the question
    - "pred": the predicted answer to the question
 - The output should be in json format and look the following:
    { score: <xxx>}
    where <xxx> is the number of stars you give to the answer. Do not add
        anything else to the answer.
Input:
```

# B  Human Rater Study Details

## B.1  Comparison of different LLMs

Figure 10 compares various LLMs and traditional metrics based on their correlation with human raters. On the SLAKE and OVQA datasets, the best-performing LLMs surpass traditional metrics in aligning with human judgments. However, on the MIMIC dataset, traditional metrics slightly outperform the LLMs. Interestingly, models specialized for the medical domain (Biomistral and Biomistral DARE) or designed as evaluators (Prometheus) do not demonstrate any advantage and even rank among the worst-performing models. On average, Gemma, Qwen, and Mistral v0.3 achieve the highest correlation with human judgments, with Gemma notably ranking in the top three across all datasets. Among the traditional metrics, the F1 score exhibits the best mean performance across datasets.

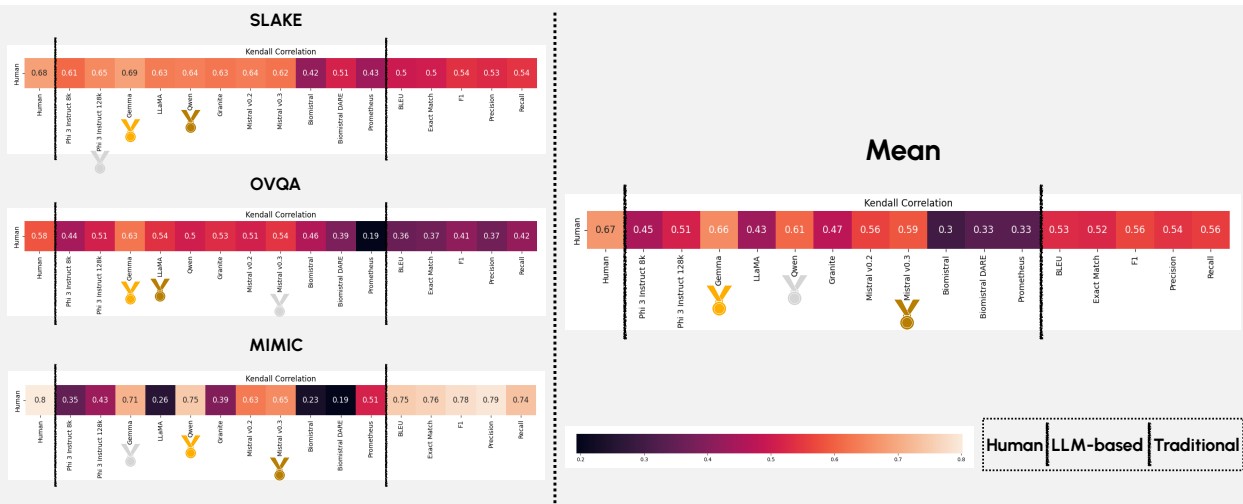

Figure 10: **Comparison of various LLMs and Traditional Metrics in the Human Rater Study**. Human interrater correlation is calculated between five human raters. Shown are the Kendall correlation between the human rating and LLM metrics as well as the traditional metrics. Gold, silver, and bronze medal show the best LLMs on the respective datasets. Left side shows the individual datasets and right side the mean over all datasets.

## B.2 Qualitative Results of the Human Rater Study

Figure 11 shows qualitative results for the human rater study. Thereby, Figure 11a shows failures of the traditional metric on the OVQA dataset, where the LLM metrics clearly outperform the traditional metrics in terms of correlation to humans. In contrast Figure 11b show failures of Gemma on the MIMIC dataset, where the traditional metrics show a slightly higher correlation with human judgment.

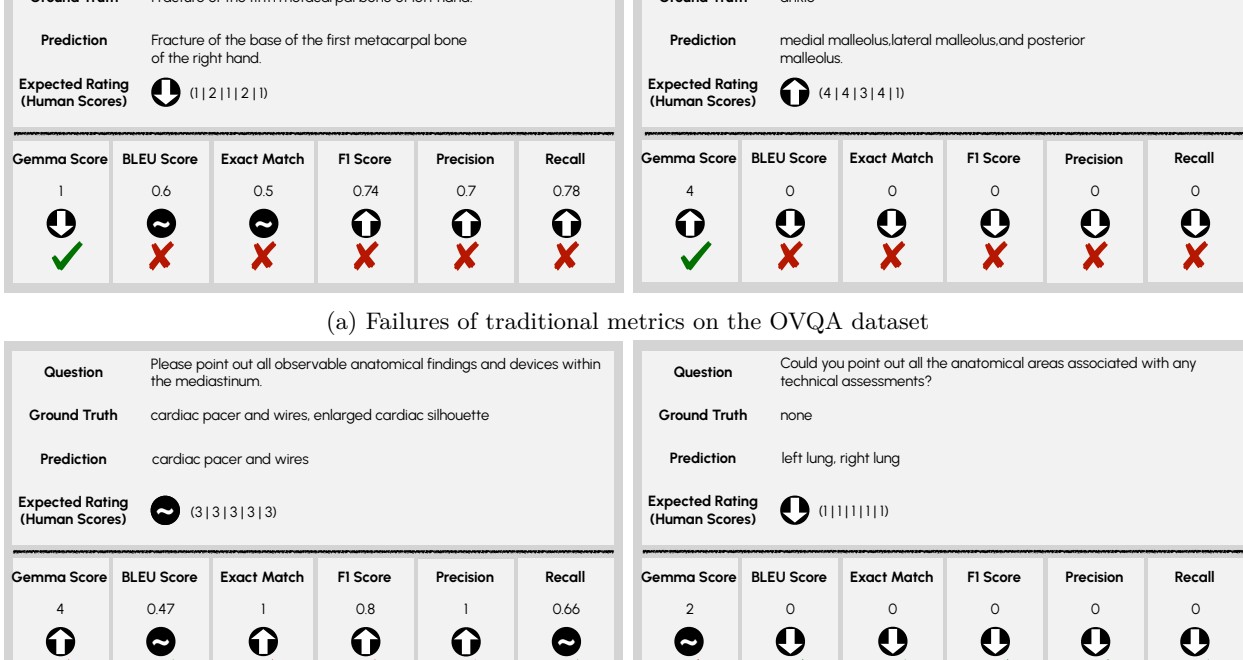

(a) Failures of traditional metrics on the OVQA dataset

(b) Failures of Gemma on the MIMIC dataset.

Figure 11: **Qualitative Results of the Human Rater Study**. For each sample, the question, ground truth, prediction, and expected score by human ratings are shown. On the bottom, for each automated metric, the absolute value is shown with an indication if it is high or low in the metrics range and an indication if the expectations from the human ratings are met. Gemma scores range from 1-5 and traditional metrics from 0-1.

## B.3 Detailed Quantitative Results of the Human Rater Study

The following figures show detailed results of the human rater study. Figure 12, Figure 13, and Figure 14 show scatter plots with the correlation between the human ratings and the other metrics. Figure 15 shows detailed correlation results, including the correlation between Gemma and the other metrics.

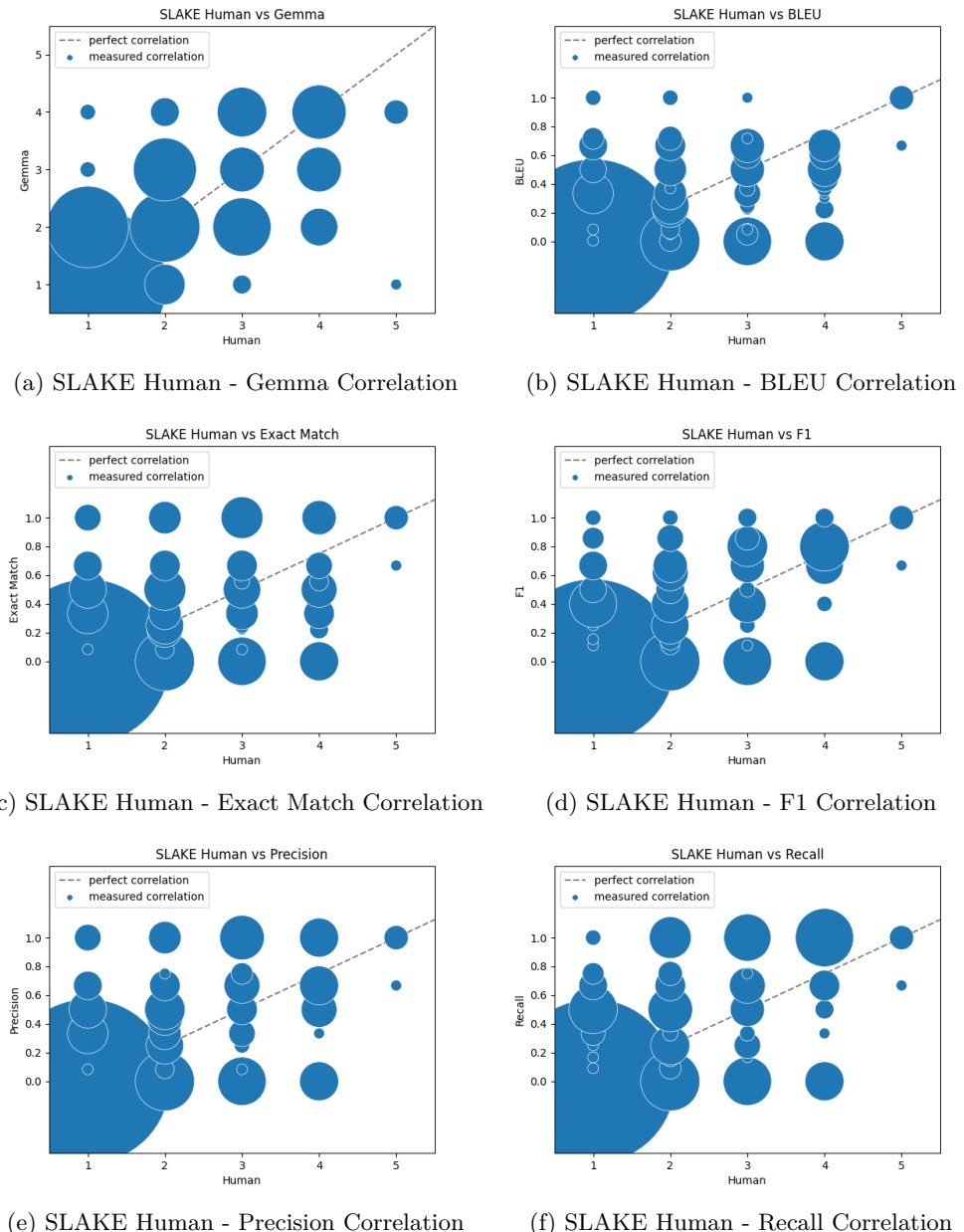

(a) SLAKE Human - Gemma Correlation

(b) SLAKE Human - BLEU Correlation

(c) SLAKE Human - Exact Match Correlation

(d) SLAKE Human - F1 Correlation

(e) SLAKE Human - Precision Correlation

(f) SLAKE Human - Recall Correlation

Figure 12: Scatter plots showing the correlation between the human ratings and respective other metrics on the SLAKE dataset. Size of the dots indicates the number of ratings that correspond to that point.

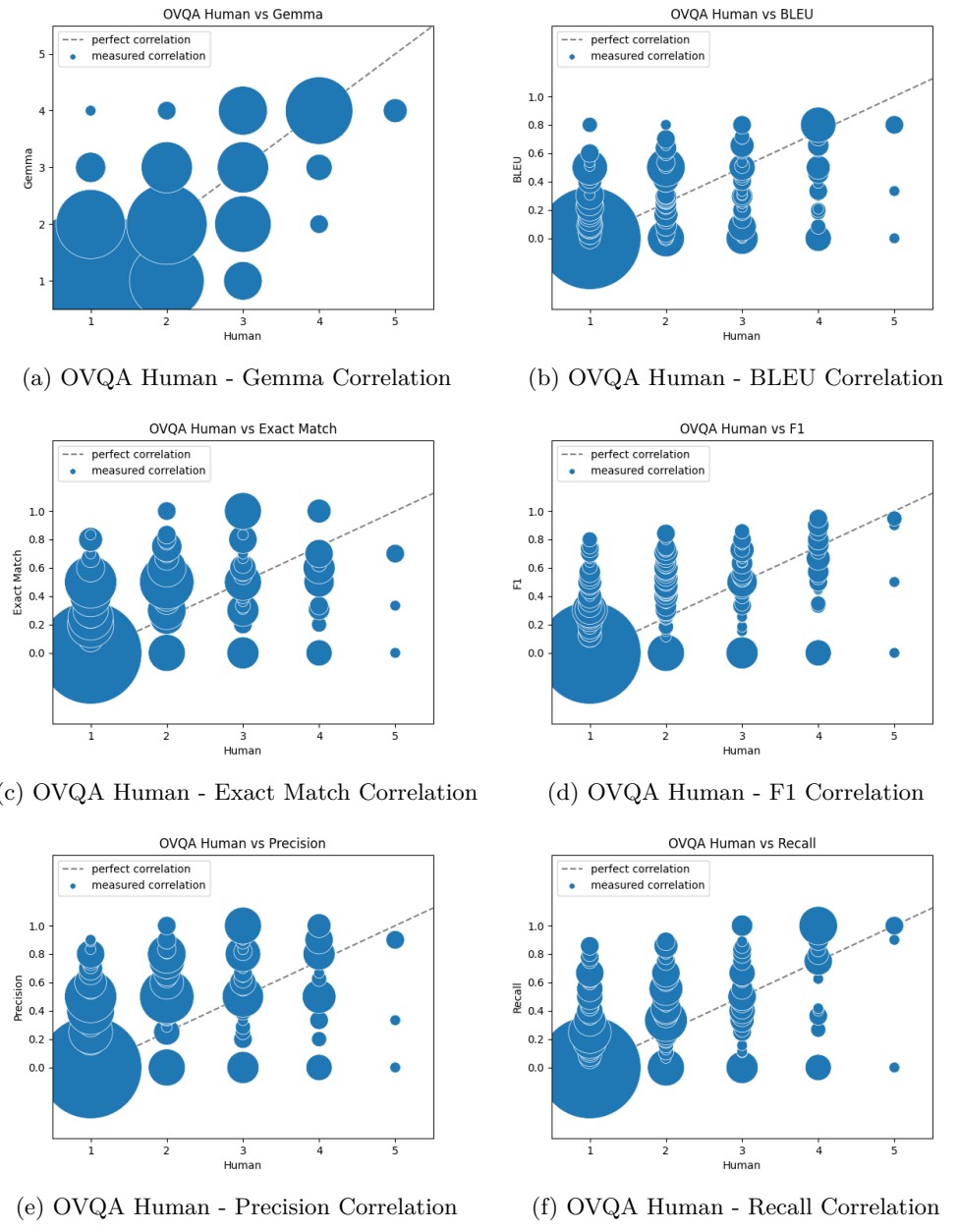

(a) OVQA Human - Gemma Correlation

(b) OVQA Human - BLEU Correlation

(c) OVQA Human - Exact Match Correlation

(d) OVQA Human - F1 Correlation

(e) OVQA Human - Precision Correlation

(f) OVQA Human - Recall Correlation

Figure 13: Scatter plots showing the correlation between the human ratings and respective other metrics on the OVQA dataset. Size of the dots indicates the number of ratings that correspond to that point.

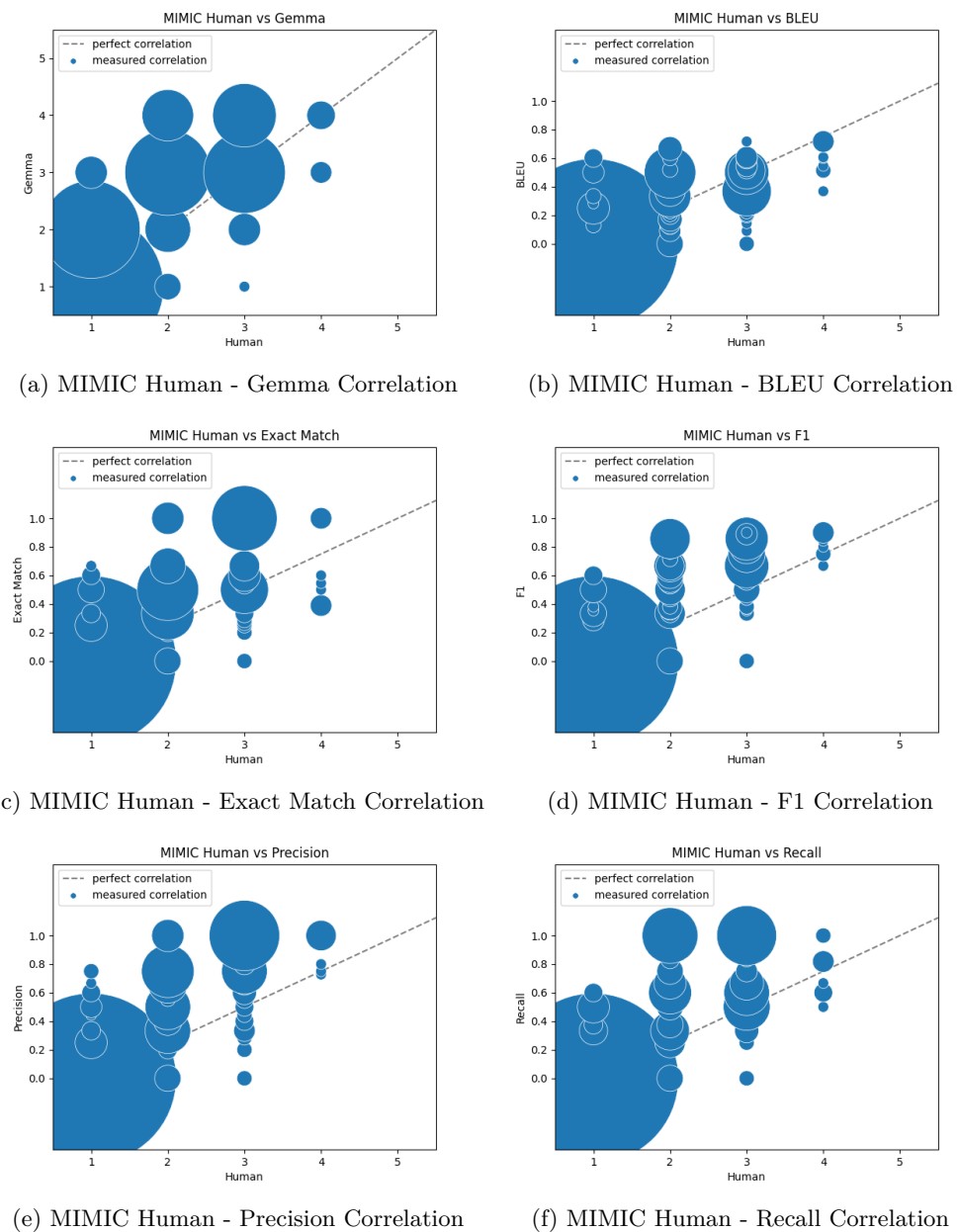

Figure 14: Scatter plots showing the correlation between the human ratings and respective other metrics on the MIMIC dataset. Size of the dots indicates the number of ratings that correspond to that point.

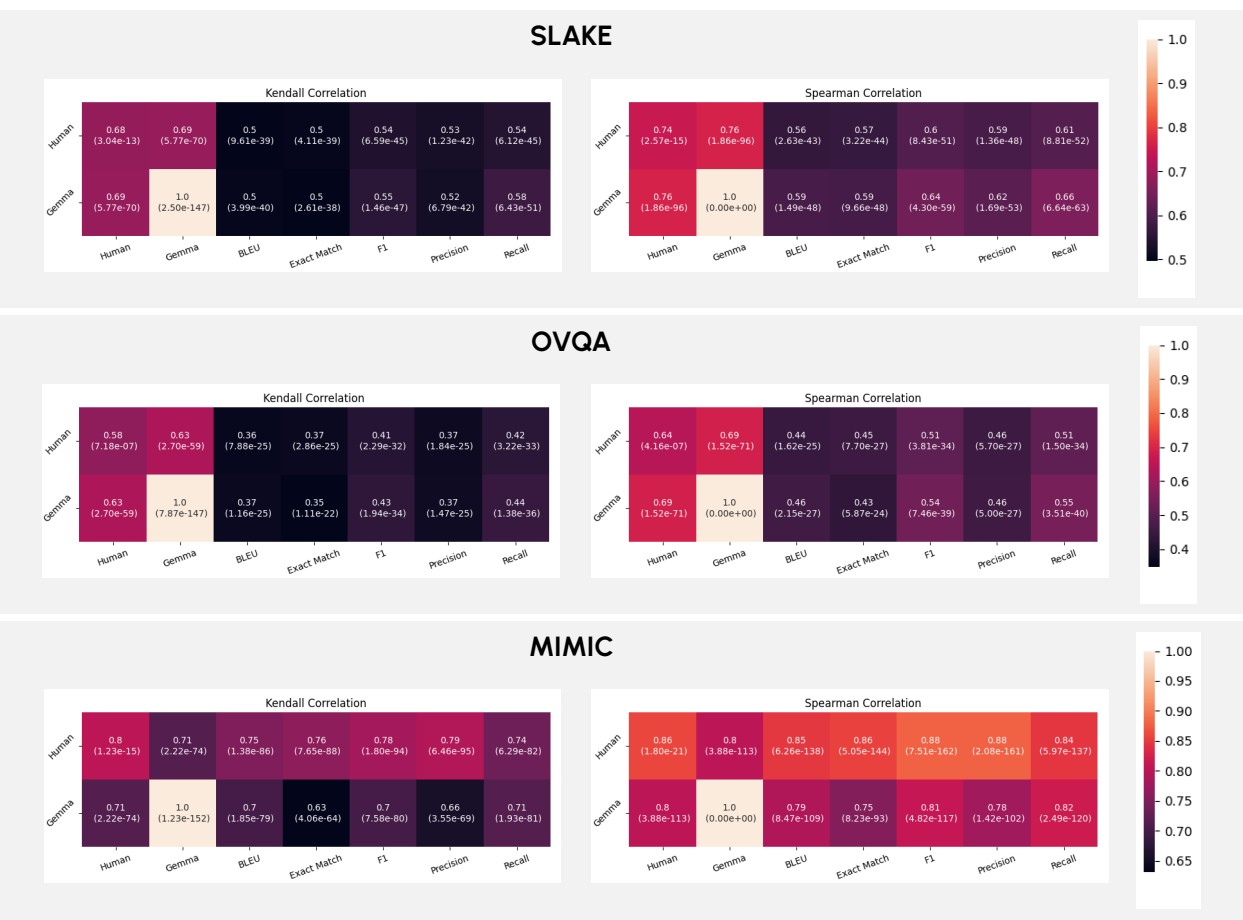

Figure 15: **Extended Results of the Human Rater Study**. Human interrater correlation is calculated between five human raters. Shown are the Kendall and Spearman correlation between the human rating and all metrics as well as the correlation between Gemma and the traditional metrics.

# C   Robustness Study Details

## C.1   Dataset Details

### C.1.1   SLAKE

The SLAKE dataset (Liu et al. (2021)) is a bilingual radiological VQA dataset, containing English and Chinese questions. We use the English subset of the SLAKE dataset. The dataset is composed of MRI, CT, and X-ray images. All images are 2D, so for the MRI and CT images, single slices are extracted. For each question, metadata information about the location, the modality, and the content is provided. Overall, the images are split into 5 different body locations, 11 different content types (question types), and the mentioned three modalities.

The exact sizes of the dataset splits are listed in Table 1. Note that for the modality shift, we merged the test set with the OoD cases from the training set, since the images are distinct, and thus, the same image cannot appear in the training and test set. As this is not the case for the question type shift, we only use the OoD cases from the test set here.

Table 1: Size of the SLAKE dataset for the different splits.

| Split | i.i.d./OoD/all | # Cases |
|---|---|---|
| **Whole Dataset** | | |
| Train | all | 4866 |
| Validate | all | 1043 |
| **Modality Shift (OoD: X-Ray)** | | |
| Train | i.i.d. | 3448 |
| Test | i.i.d. | 689 |
| Test | OoD | 1779 |
| **Question Type Shift (OoD: Size)** | | |
| Train | i.i.d. | 4581 |
| Test | i.i.d. | 994 |
| Test | OoD | 56 |

### C.1.2   OVQA

The OVQA dataset (Huang et al. (2022)) is an orthopedic VQA dataset, containing CT and X-Ray images. All images are 2D, so for the CT images, either a 3D rendering is shown as a 2D image or a single plane. For each question, metadata information is provided about the imaged organ (like the "location" in the SLAKE dataset), and the question type (like the "content" in SLAKE). The dataset contains 6 different question types and 4 different body parts.

The exact sizes of the dataset splits are listed in Table 2. We removed closed-ended questions with more than two categories to choose from and closed-ended questions where the categories to answer were not exactly contained in the question. As for the SLAKE dataset, we merged the questions from the training set to the OoD test set for the organ shift, but not for the question type shift.

Table 2: Size of the OVQA dataset for the different splits.

| Split | i.i.d./OoD/all | # Cases |
|---|---|---|
| **Whole Dataset** | | |
| Train | all | 13492 |
| Validate | all | 1645 |
| **Organ Shift (OoD: Leg)** | | |
| Train | i.i.d. | 8755 |
| Test | i.i.d. | 1044 |
| Test | OoD | 5350 |
| **Question Type Shift (OoD: Organ System)** | | |
| Train | i.i.d. | 11924 |
| Test | i.i.d. | 1420 |
| Test | OoD | 237 |

### C.1.3   MIMIC-CXR-VQA

The MIMIC-CXR-VQA dataset (Bae et al. (2023)) is a chest X-ray dataset, which is built based on the MIMIC-CXR dataset (Johnson et al. (2019)), the MIMIC-IV dataset (Johnson et al. (2023)), and the Chest ImaGenome dataset (Wu et al. (2021)). For each question, the semantic type is specified. Three different semantic types are specified, which are "choose", "query", and "verify". For "choose", the task is to choose between two options provided in the answer, but also both or none of the options can be correct. For "query", the task is to list all the categories that match the questions, e.g. all anatomical findings. Lastly, "verify" are yes/no questions. All the questions can be answered based on a fixed set of classes, where the dataset overall contains 110 answer labels. The answers are given as a list of the correct classes. We preprocess the questions differently, based on their semantic type: For the "choose" questions, whenever the list of answers contains both options, we change the answer to "both", and whenever the list of answers is empty, we change the answer to "none". For the "query" questions, we concatenate the list of answers to one string, with the answer labels being comma-separated. For the "verify" questions, we do not apply any specific preprocessing. The information for the patient's gender, ethnicity, and age is taken from the MIMIC-IV dataset. Whenever the metadata information of a subject ID is not unique, we set it to "none". In the respective shifts, we exclude questions where the corresponding metadata field is not known, which includes all fields with "none", and for the ethnicity shift also the value "unknown/other". The exact sizes of the dataset splits are listed in Table 3.

Table 3: Size of the MIMIC dataset for the different splits.

| Split | i.i.d./OoD/all | # Cases |
|---|---|---|
| **Whole Dataset** | | |
| Train | all | 290031 |
| Validate | all | 73567 |
| **Gender Shift (OoD: Female)** | | |
| Train | i.i.d. | 147790 |
| Test | i.i.d. | 7277 |
| Test | OoD | 6120 |
| **Ethnicity Shift (OoD: Non-white)** | | |
| Train | i.i.d. | 171593 |
| Test | i.i.d. | 8101 |
| Test | OoD | 3713 |
| **Age Shift (OoD: Young)** | | |
| Train | i.i.d. | 155941 |
| Test | i.i.d. | 6686 |
| Test | OoD | 2076 |

### C.1.4   Ratio of unique questions in the datasets

Table 4: Ratio of unique questions in the datasets

| | | Overall | Unique | Ratio |
|---|---|---|---|---|
| | **MIMIC** | 290031 | 132387 | 0.46 |
| **Train** | **SLAKE** | 4866 | 579 | 0.12 |
| | **OVQA** | 13492 | 960 | 0.07 |
| | **MIMIC** | 73567 | 31148 | 0.42 |
| **Val** | **SLAKE** | 1043 | 314 | 0.3 |
| | **OVQA** | 1645 | 266 | 0.16 |
| | **MIMIC** | 13793 | 7565 | 0.55 |
| **Test** | **SLAKE** | 1050 | 313 | 0.3 |
| | **OVQA** | 1657 | 335 | 0.2 |

## C.2 Hyperparameter Search

We performed several hyperparameter sweeps for each dataset and PEFT method in order to find suitable setups for the experiments in the FT robustness study. For the hyperparameter sweeps, we trained on the whole training set for each dataset and PEFT method and ran inference on the validation set. Training ran for 3 epochs and 3 seeds for each experiment.

### C.2.1 Prompt Tuning

For prompt tuning, we performed the following hyperparameter sweeps:

- Number of tokens: $[40, 60, 80, 100]$
- Learning rate: $[3e - 2, 3e - 1]$

The results for SLAKE can be found in Table 5, for OVQA in Table 6, and for MIMIC in Table 7.

Table 5: Hyperparameter sweep for prompt tuning on the SLAKE dataset. Selected hyperparameters for the final FT robustness study are highlighted. Mean and standard deviation are reported for three seeds.

(a) Hyperparameter sweep for the medical base model.

| # Tokens | Learning Rate | Closed-Ended (Gemma Accuracy) | Open-Ended (Gemma Score) |
|---|---|---|---|
| 40 | 3e-2 | 0.79 +/- 0.02 | 4.21 +/- 0.03 |
| 40 | 3e-1 | 0.8 +/- 0.02 | 4.24 +/- 0.06 |
| 60 | 3e-2 | 0.76 +/- 0.04 | 4.22 +/- 0.03 |
| 60 | 3e-1 | 0.81 +/- 0.01 | 4.22 +/- 0.06 |
| 80 | 3e-2 | 0.78 +/- 0.05 | 4.21 +/- 0.02 |
| 80 | 3e-1 | 0.82 +/- 0.01 | 4.23 +/- 0.02 |
| 100 | 3e-2 | 0.77 +/- 0.06 | 4.21 +/- 0.05 |
| 100 | 3e-1 | 0.81 +/- 0.0 | 4.23 +/- 0.04 |

(b) Hyperparameter sweep for the non-medical base model.

| # Tokens | Learning Rate | Closed-Ended (Gemma Accuracy) | Open-Ended (Gemma Score) |
|---|---|---|---|
| 40 | 3e-2 | 0.76 +/- 0.04 | 4.15 +/- 0.02 |
| 40 | 3e-1 | 0.79 +/- 0.01 | 4.18 +/- 0.02 |
| 60 | 3e-2 | 0.77 +/- 0.0 | 4.13 +/- 0.02 |
| 60 | 3e-1 | 0.79 +/- 0.01 | 4.17 +/- 0.02 |
| 80 | 3e-2 | 0.76 +/- 0.01 | 4.14 +/- 0.05 |
| 80 | 3e-1 | 0.78 +/- 0.01 | 4.17 +/- 0.01 |
| 100 | 3e-2 | 0.77 +/- 0.02 | 4.17 +/- 0.02 |
| 100 | 3e-1 | 0.79 +/- 0.02 | 4.17 +/- 0.06 |

Table 6: Hyperparameter sweep for prompt tuning on the OVQA dataset. Selected hyperparameters for the final FT robustness study are highlighted. Mean and standard deviation are reported for three seeds.

(a) Hyperparameter sweep for the medical base model.

| # Tokens | Learning Rate | Closed-Ended (Gemma Accuracy) | Open-Ended (Gemma Score) |
|---|---|---|---|
| 40 | 3e-2 | 0.85 +/- 0.0 | 3.05 +/- 0.05 |
| 40 | 3e-1 | 0.85 +/- 0.01 | 3.05 +/- 0.04 |
| 60 | 3e-2 | 0.85 +/- 0.01 | 3.06 +/- 0.03 |
| 60 | 3e-1 | 0.85 +/- 0.0 | 3.05 +/- 0.04 |
| 80 | 3e-2 | 0.82 +/- 0.03 | 3.05 +/- 0.05 |
| 80 | 3e-1 | 0.84 +/- 0.01 | 3.07 +/- 0.03 |
| 100 | 3e-2 | 0.83 +/- 0.02 | 3.1 +/- 0.03 |
| 100 | 3e-1 | 0.85 +/- 0.0 | 3.08 +/- 0.02 |

(b) Hyperparameter sweep for the non-medical base model.

| # Tokens | Learning Rate | Closed-Ended (Gemma Accuracy) | Open-Ended (Gemma Score) |
|---|---|---|---|
| 40 | 3e-2 | 0.8 +/- 0.03 | 2.97 +/- 0.02 |
| 40 | 3e-1 | 0.84 +/- 0.01 | 2.98 +/- 0.03 |
| 60 | 3e-2 | 0.81 +/- 0.02 | 2.97 +/- 0.06 |
| 60 | 3e-1 | 0.83 +/- 0.0 | 3.01 +/- 0.05 |
| 80 | 3e-2 | 0.81 +/- 0.03 | 3.04 +/- 0.05 |
| 80 | 3e-1 | 0.84 +/- 0.0 | 3.0 +/- 0.05 |
| 100 | 3e-2 | 0.81 +/- 0.01 | 3.0 +/- 0.06 |
| 100 | 3e-1 | 0.84 +/- 0.01 | 3.03 +/- 0.07 |

Table 7: Hyperparameter sweep for prompt tuning on the MIMIC dataset. Selected hyperparameters for the final FT robustness study are highlighted. Mean and standard deviation are reported for three seeds.

(a) Hyperparameter sweep for the medical base model.

| # Tokens | Learning Rate | Closed-Ended (Gemma Accuracy) | Open-Ended (Gemma Score) |
|---|---|---|---|
| 40 | 3e-2 | 0.67 +/- 0.02 | 3.21 +/- 0.03 |
| 40 | 3e-1 | 0.67 +/- 0.01 | 3.2 +/- 0.05 |
| 60 | 3e-2 | 0.68 +/- 0.01 | 3.18 +/- 0.01 |
| 60 | 3e-1 | 0.69 +/- 0.01 | 3.24 +/- 0.02 |
| 80 | 3e-2 | 0.66 +/- 0.05 | 3.21 +/- 0.02 |
| 80 | 3e-1 | 0.69 +/- 0.02 | 3.21 +/- 0.03 |
| 100 | 3e-2 | 0.68 +/- 0.02 | 3.23 +/- 0.03 |
| 100 | 3e-1 | 0.67 +/- 0.01 | 3.21 +/- 0.03 |

(b) Hyperparameter sweep for the non-medical base model.

| # Tokens | Learning Rate | Closed-Ended (Gemma Accuracy) | Open-Ended (Gemma Score) |
|---|---|---|---|
| 40 | 3e-2 | 0.69 +/- 0.0 | 3.16 +/- 0.03 |
| 40 | 3e-1 | 0.69 +/- 0.0 | 3.18 +/- 0.03 |
| 60 | 3e-2 | 0.69 +/- 0.0 | 3.22 +/- 0.04 |
| 60 | 3e-1 | 0.68 +/- 0.03 | 3.19 +/- 0.02 |
| 80 | 3e-2 | 0.68 +/- 0.01 | 3.2 +/- 0.01 |
| 80 | 3e-1 | 0.7 +/- 0.0 | 3.19 +/- 0.04 |
| 100 | 3e-2 | 0.69 +/- 0.01 | 3.21 +/- 0.04 |
| 100 | 3e-1 | 0.68 +/- 0.01 | 3.17 +/- 0.02 |

### C.2.2 LoRA

For LoRA, we performed the following hyperparameter sweeps:

- Rank: $[16, 32, 64, 128, 256]$
- Learning rate: $[3e-5, 3e-4]$

$\alpha$ is set to $2 \times$ Rank. The results for SLAKE can be found in Table 8, for OVQA in Table 9, and for MIMIC in Table 10. Note that some of the hyperparameter configurations led to instabilities during training loss, indicated by "NaN".

Table 8: Hyperparameter sweep for LoRA on the SLAKE dataset. Selected hyperparameters for the final FT robustness study are highlighted. Mean and standard deviation are reported for three seeds. Rows with "NaN" showed instabilities in the loss during training.

(a) Hyperparameter sweep for the medical base model.

| Rank | Learning Rate | Closed-Ended (Gemma Accuracy) | Open-Ended (Gemma Score) |
|---|---|---|---|
| 16 | 3e-5 | 0.83 +/- 0.01 | 4.29 +/- 0.02 |
| 16 | 3e-4 | 0.82 +/- 0.01 | 4.27 +/- 0.04 |
| 32 | 3e-5 | 0.85 +/- 0.01 | 4.31 +/- 0.03 |
| 32 | 3e-4 | 0.73 +/- 0.07 | 4.25 +/- 0.01 |
| 64 | 3e-5 | 0.84 +/- 0.01 | 4.33 +/- 0.04 |
| 64 | 3e-4 | 0.51 +/- 0.07 | 2.92 +/- 1.41 |
| 128 | 3e-5 | 0.84 +/- 0.01 | 4.34 +/- 0.02 |
| 128 | 3e-4 | NaN | NaN |
| 256 | 3e-5 | 0.83 +/- 0.01 | 4.31 +/- 0.01 |
| 256 | 3e-4 | NaN | NaN |

(b) Hyperparameter sweep for the non-medical base model.

| Rank | Learning Rate | Closed-Ended (Gemma Accuracy) | Open-Ended (Gemma Score) |
|---|---|---|---|
| 16 | 3e-5 | 0.84 +/- 0.01 | 4.25 +/- 0.02 |
| 16 | 3e-4 | 0.82 +/- 0.01 | 4.26 +/- 0.02 |
| 32 | 3e-5 | 0.84 +/- 0.0 | 4.33 +/- 0.02 |
| 32 | 3e-4 | 0.78 +/- 0.01 | 4.2 +/- 0.05 |
| 64 | 3e-5 | 0.86 +/- 0.01 | 4.34 +/- 0.02 |
| 64 | 3e-4 | 0.45 +/- 0.09 | 1.86 +/- 0.78 |
| 128 | 3e-5 | 0.86 +/- 0.01 | 4.32 +/- 0.04 |
| 128 | 3e-4 | NaN | NaN |
| 256 | 3e-5 | 0.84 +/- 0.01 | 4.31 +/- 0.03 |
| 256 | 3e-4 | NaN | NaN |

Table 9: Hyperparameter sweep for LoRA on the OVQA dataset. Selected hyperparameters for the final FT robustness study are highlighted. Mean and standard deviation are reported for three seeds. Rows with "NaN" showed instabilities in the loss during training.

(a) Hyperparameter sweep for the medical base model.

| Rank | Learning Rate | Closed-Ended (Gemma Accuracy) | Open-Ended (Gemma Score) |
|------|---------------|-------------------------------|--------------------------|
| 16   | 3e-5          | 0.84 +/- 0.0                  | 3.13 +/- 0.06            |
| 16   | 3e-4          | 0.83 +/- 0.02                 | 3.18 +/- 0.02            |
| 32   | 3e-5          | 0.85 +/- 0.0                  | 3.15 +/- 0.01            |
| 32   | 3e-4          | 0.82 +/- 0.01                 | 3.08 +/- 0.04            |
| 64   | 3e-5          | 0.85 +/- 0.0                  | 3.2 +/- 0.02             |
| 64   | 3e-4          | 0.66 +/- 0.01                 | 2.03 +/- 0.13            |
| 128  | 3e-5          | 0.85 +/- 0.0                  | 3.18 +/- 0.03            |
| 128  | 3e-4          | NaN                           | NaN                      |
| 256  | 3e-5          | 0.85 +/- 0.0                  | 3.18 +/- 0.02            |
| 256  | 3e-4          | NaN                           | NaN                      |

(b) Hyperparameter sweep for the non-medical base model.

| Rank | Learning Rate | Closed-Ended (Gemma Accuracy) | Open-Ended (Gemma Score) |
|------|---------------|-------------------------------|--------------------------|
| 16   | 3e-5          | 0.84 +/- 0.0                  | 3.08 +/- 0.03            |
| 16   | 3e-4          | 0.81 +/- 0.02                 | 3.17 +/- 0.02            |
| 32   | 3e-5          | 0.84 +/- 0.01                 | 3.12 +/- 0.04            |
| 32   | 3e-4          | 0.81 +/- 0.03                 | 3.07 +/- 0.03            |
| 64   | 3e-5          | 0.84 +/- 0.01                 | 3.15 +/- 0.01            |
| 64   | 3e-4          | 0.61 +/- 0.07                 | 2.12 +/- 0.11            |
| 128  | 3e-5          | 0.84 +/- 0.0                  | 3.2 +/- 0.04             |
| 128  | 3e-4          | NaN                           | NaN                      |
| 256  | 3e-5          | 0.84 +/- 0.01                 | 3.2 +/- 0.05             |
| 256  | 3e-4          | NaN                           | NaN                      |

Table 10: Hyperparameter sweep for LoRA on the MIMIC dataset. Selected hyperparameters for the final FT robustness study are highlighted. Mean and standard deviation are reported for three seeds. Rows with "NaN" showed instabilities in the loss during training.

(a) Hyperparameter sweep for the medical base model.

| Rank | Learning Rate | Closed-Ended (Gemma Accuracy) | Open-Ended (Gemma Score) |
|------|---------------|-------------------------------|--------------------------|
| 16   | 3e-5          | 0.71 +/- 0.01                 | 3.33 +/- 0.02            |
| 16   | 3e-4          | 0.68 +/- 0.01                 | 3.23 +/- 0.05            |
| 32   | 3e-5          | 0.71 +/- 0.0                  | 3.36 +/- 0.01            |
| 32   | 3e-4          | 0.42 +/- 0.18                 | 2.43 +/- 0.07            |
| 64   | 3e-5          | 0.71 +/- 0.01                 | 3.35 +/- 0.02            |
| 64   | 3e-4          | NaN                           | NaN                      |
| 128  | 3e-5          | 0.71 +/- 0.0                  | 3.37 +/- 0.02            |
| 128  | 3e-4          | NaN                           | NaN                      |
| 256  | 3e-5          | NaN                           | NaN                      |
| 256  | 3e-4          | NaN                           | NaN                      |

(b) Hyperparameter sweep for the non-medical base model.

| Rank | Learning Rate | Closed-Ended (Gemma Accuracy) | Open-Ended (Gemma Score) |
|------|---------------|-------------------------------|--------------------------|
| 16   | 3e-5          | 0.7 +/- 0.01                  | 3.28 +/- 0.03            |
| 16   | 3e-4          | 0.68 +/- 0.01                 | 3.25 +/- 0.05            |
| 32   | 3e-5          | 0.7 +/- 0.0                   | 3.34 +/- 0.04            |
| 32   | 3e-4          | 0.44 +/- 0.07                 | 2.43 +/- 0.05            |
| 64   | 3e-5          | 0.71 +/- 0.01                 | 3.34 +/- 0.02            |
| 64   | 3e-4          | NaN                           | NaN                      |
| 128  | 3e-5          | 0.7 +/- 0.0                   | 3.36 +/- 0.03            |
| 128  | 3e-4          | NaN                           | NaN                      |
| 256  | 3e-5          | 0.7 +/- 0.02                  | 3.37 +/- 0.02            |
| 256  | 3e-4          | NaN                           | NaN                      |

## C.3  (IA)$^3$

For (`IA`)$^3$, we performed the following hyperparameter sweeps:

- Learning rate: $[3e-3, 3e-2, 3e-1]$

The results for SLAKE can be found in Table 11, for OVQA in Table 12, and for MIMIC in Table 13.

Table 11: Hyperparameter sweep for (`IA`)$^3$ on the SLAKE dataset. Selected hyperparameters for the final FT robustness study are highlighted. Mean and standard deviation are reported for three seeds.

(a) Hyperparameter sweep for the medical base model.

| Learning Rate | Closed-Ended (Gemma Accuracy) | Open-Ended (Gemma Score) |
|---|---|---|
| lr3e-3 | 0.63 +/- 0.02 | 3.72 +/- 0.01 |
| lr3e-2 | 0.83 +/- 0.01 | 4.32 +/- 0.03 |
| lr3e-1 | 0.65 +/- 0.01 | 4.24 +/- 0.04 |

(b) Hyperparameter sweep for the non-medical base model.

| Learning Rate | Closed-Ended (Gemma Accuracy) | Open-Ended (Gemma Score) |
|---|---|---|
| lr3e-3 | 0.75 +/- 0.01 | 3.61 +/- 0.01 |
| lr3e-2 | 0.86 +/- 0.01 | 4.27 +/- 0.02 |
| lr3e-1 | 0.66 +/- 0.03 | 4.18 +/- 0.04 |

Table 12: Hyperparameter sweep for (`IA`)$^3$ on the OVQA dataset. Selected hyperparameters for the final FT robustness study are highlighted. Mean and standard deviation are reported for three seeds.

(a) Hyperparameter sweep for the medical base model.

| Learning Rate | Closed-Ended (Gemma Accuracy) | Open-Ended (Gemma Score) |
|---|---|---|
| lr3e-3 | 0.75 +/- 0.01 | 2.91 +/- 0.04 |
| lr3e-2 | 0.84 +/- 0.0 | 3.15 +/- 0.03 |
| lr3e-1 | 0.8 +/- 0.04 | 3.08 +/- 0.04 |

(b) Hyperparameter sweep for the non-medical base model.

| Learning Rate | Closed-Ended (Gemma Accuracy) | Open-Ended (Gemma Score) |
|---|---|---|
| lr3e-3 | 0.75 +/- 0.02 | 2.65 +/- 0.03 |
| lr3e-2 | 0.84 +/- 0.0 | 3.1 +/- 0.05 |
| lr3e-1 | 0.8 +/- 0.02 | 3.05 +/- 0.03 |

Table 13: Hyperparameter sweep for (`IA`)$^3$ on the MIMIC dataset. Selected hyperparameters for the final FT robustness study are highlighted. Mean and standard deviation are reported for three seeds.

(a) Hyperparameter sweep for the medical base model.

| Learning Rate | Closed-Ended (Gemma Accuracy) | Open-Ended (Gemma Score) |
|---|---|---|
| lr3e-3 | 0.53 +/- 0.01 | 2.92 +/- 0.01 |
| lr3e-2 | 0.7 +/- 0.01 | 3.33 +/- 0.02 |
| lr3e-1 | 0.61 +/- 0.05 | 3.1 +/- 0.03 |

(b) Hyperparameter sweep for the non-medical base model.

| Learning Rate | Closed-Ended (Gemma Accuracy) | Open-Ended (Gemma Score) |
|---|---|---|
| lr3e-3 | 0.6 +/- 0.01 | 2.7 +/- 0.02 |
| lr3e-2 | 0.7 +/- 0.0 | 3.33 +/- 0.02 |
| lr3e-1 | 0.6 +/- 0.11 | 3.0 +/- 0.06 |

### C.4 Robustness Results for the Non-Medical Base Model

The robustness study results for the non-medical base model are presented in Figure 16 and Figure 17, corresponding to Figure 7 and Figure 8 for the medical base model. The main trends analyzed in section 3.4 for the medical model also apply to the non-medical base model, as indicated by Figure 6, which suggests similar behavior across the two models.

While the overall trends are consistent, some minor differences are observed. For example, LoRA is no longer an outlier for closed-ended questions in SLAKE. However, a similar trend appears for the non-medical model on OVQA, where LoRA shows a lower RR compared to Prompt and $(\texttt{IA})^3$, primarily due to the question type shift. Additionally, while the question type shift continues to cause significant failures, this effect is now limited to the OVQA dataset, affecting both open-ended and closed-ended questions. Finally, while the trends for full FT are similar on the SLAKE and the OVQA dataset between the medical and the non-medical base model, the non-medical base model performs worse on the MIMIC dataset than its medical counterpart.

Despite these differences, the results confirm the main insights from the robustness study: None of the methods consistently outperforms others in terms of robustness, and robustness trends are more consistent across FT methods than across shifts. LoRA remains the best-performing PEFT method overall on the i.i.d. data.

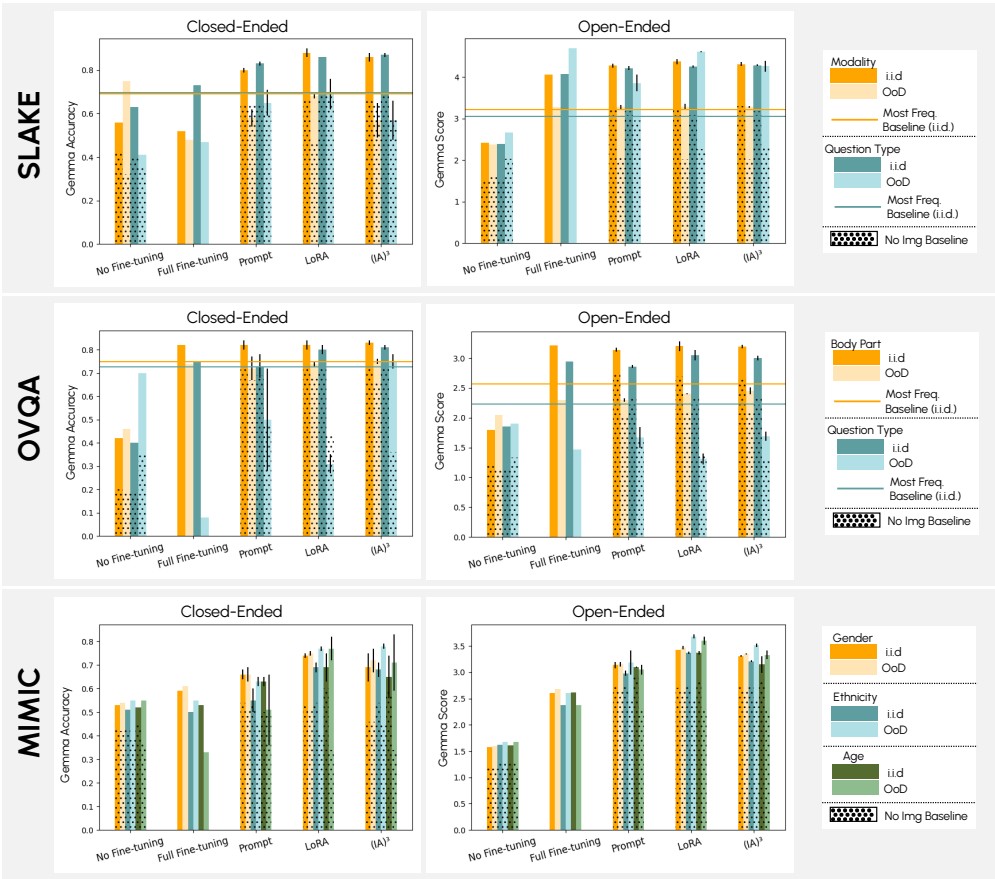

Figure 16: **Results of the FT Robustness Study on the i.i.d. and OoD Test Set for the Non-Medical Base Model**. Reported results show the mean over three seeds (exception: no FT, full FT) with the standard deviation for the non-baselines. Gemma Accuracy refers to the accuracy being rated by Gemma. For MIMIC, the *most frequent* sanity baseline can not be calculated as too few questions match the training set.

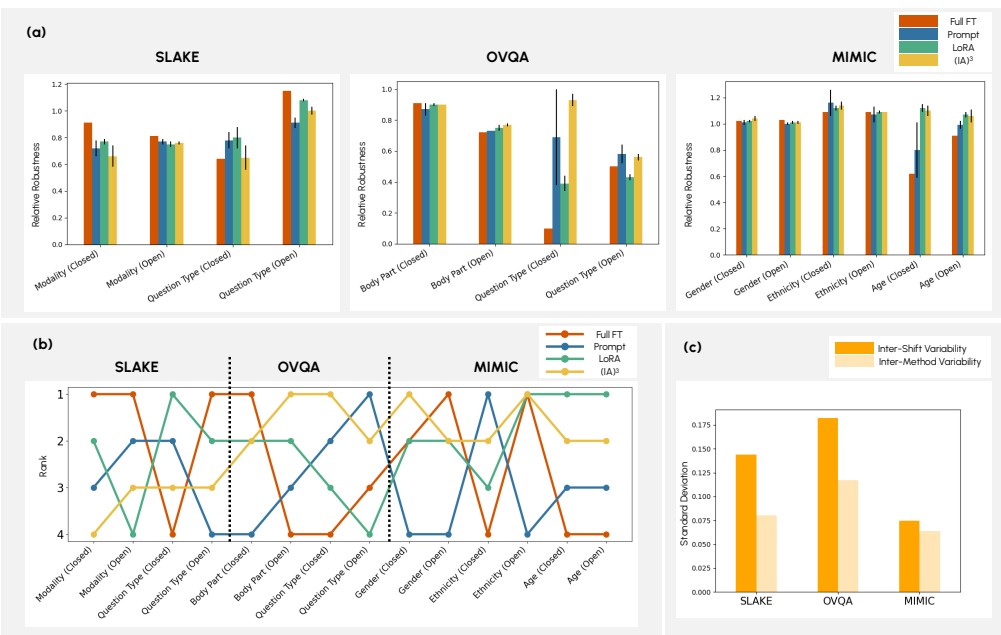

Figure 17: **Results of the FT Robustness Study Focusing on the Relative Robustness (RR) for the Non-Medical Base Model**. Since the study focuses on comparing the FT method and our definition of OoD holds for these methods, the *no FT* baseline is excluded. (a) RR on the three datasets for the methods. Results show the mean and standard deviation over three seeds (exception: full FT). (b) RR Ranking of all FT methods. (c) Standard deviation between shifts vs. standard deviation between FT methods.

## C.5 Detailed Results of the Robustness Study

Tables 14-19 show the detailed results of the robustness study. Further, Figure 18 shows the inter-method and inter-shift variability of the different PEFT methods for the medical base model, so not including full FT.

Table 14: **Robustness Results on the SLAKE Dataset.** Results with ± indicate the mean and standard deviation over three seeds. Note that the most frequent baseline can only be calculated for the i.i.d. set as for OoD too few questions match the training set. RR: Relative Robustness.

(a) Results for the medical base model.

| | Modality Shift OoD: X-Ray | | | | | | Question Type Shift OoD: Size | | | | | |
| | Closed-Ended | | | Open-Ended | | | Closed-Ended | | | Open-Ended | | |
| | i.i.d. | OoD | RR | i.i.d. | OoD | RR | i.i.d. | OoD | RR | i.i.d. | OoD | RR |
|---|---|---|---|---|---|---|---|---|---|---|---|---|
| No Finetuning | 0.6 | 0.31 | 0.51 | 2.57 | 2.6 | 1.01 | 0.54 | 0.71 | 1.3 | 2.53 | 3.1 | 1.23 |
| Full Finetuning | 0.57 | 0.48 | 0.84 | 4.09 | 3.25 | 0.79 | 0.56 | 0.35 | 0.63 | 4.14 | 4.38 | 1.06 |
| Prompt Tuning | 0.85 +/- 0.01 | 0.62 +/- 0.03 | 0.73 +/- 0.05 | 4.27 +/- 0.03 | 3.63 +/- 0.03 | 0.85 +/- 0.01 | 0.85 +/- 0.01 | 0.49 +/- 0.12 | 0.57 +/- 0.14 | 4.23 +/- 0.01 | 4.34 +/- 0.12 | 1.03 +/- 0.03 |
| LoRA | 0.88 +/- 0.0 | 0.45 +/- 0.04 | 0.51 +/- 0.04 | 4.36 +/- 0.06 | 3.59 +/- 0.07 | 0.82 +/- 0.03 | 0.87 +/- 0.0 | 0.39 +/- 0.07 | 0.45 +/- 0.08 | 4.29 +/- 0.02 | 4.15 +/- 0.07 | 0.97 +/- 0.01 |
| IA3 | 0.85 +/- 0.01 | 0.64 +/- 0.07 | 0.75 +/- 0.08 | 4.36 +/- 0.03 | 3.46 +/- 0.12 | 0.79 +/- 0.03 | 0.87 +/- 0.02 | 0.53 +/- 0.06 | 0.61 +/- 0.06 | 4.27 +/- 0.01 | 4.15 +/- 0.22 | 0.97 +/- 0.05 |
| Most Frequent | 0.69 | - | - | 3.25 | - | - | 0.696 | - | - | 3.1 | - | - |

(b) Results for the non-medical base model.

| | Modality Shift OoD: X-Ray | | | | | | Question Type Shift OoD: Size | | | | | |
| | Closed-Ended | | | Open-Ended | | | Closed-Ended | | | Open-Ended | | |
| | i.i.d. | OoD | RR | i.i.d. | OoD | RR | i.i.d. | OoD | RR | i.i.d. | OoD | RR |
|---|---|---|---|---|---|---|---|---|---|---|---|---|
| No Finetuning | 0.56 | 0.75 | 1.34 | 2.42 | 2.38 | 0.98 | 0.63 | 0.41 | 0.66 | 2.39 | 2.67 | 1.11 |
| Full Finetuning | 0.52 | 0.48 | 0.91 | 4.05 | 3.27 | 0.81 | 0.73 | 0.47 | 0.64 | 4.07 | 4.69 | 1.15 |
| Prompt Tuning | 0.8 +/- 0.01 | 0.58 +/- 0.04 | 0.72 +/- 0.06 | 4.27 +/- 0.05 | 3.28 +/- 0.05 | 0.77 +/- 0.02 | 0.83 +/- 0.01 | 0.65 +/- 0.06 | 0.78 +/- 0.06 | 4.21 +/- 0.05 | 3.85 +/- 0.2 | 0.91 +/- 0.04 |
| LoRA | 0.88 +/- 0.02 | 0.68 +/- 0.01 | 0.77 +/- 0.02 | 4.37 +/- 0.06 | 3.28 +/- 0.02 | 0.75 +/- 0.01 | 0.86 +/- 0.0 | 0.69 +/- 0.07 | 0.8 +/- 0.08 | 4.25 +/- 0.03 | 4.61 +/- 0.01 | 1.08 +/- 0.01 |
| IA3 | 0.86 +/- 0.02 | 0.57 +/- 0.08 | 0.66 +/- 0.08 | 4.31 +/- 0.05 | 3.28 +/- 0.02 | 0.76 +/- 0.01 | 0.87 +/- 0.01 | 0.57 +/- 0.09 | 0.65 +/- 0.09 | 4.28 +/- 0.02 | 4.26 +/- 0.13 | 1.0 +/- 0.03 |
| Most Frequent | 0.69 | - | - | 3.25 | - | - | 0.696 | - | - | 3.1 | - | - |

Table 15: **No Image Baseline on the SLAKE Dataset.** Results with ± indicate the mean and standard deviation over three seeds. The model was trained with the same methods as Table 14 just without seeing the image content. RR: Relative Robustness.

(a) Results for the medical base model.

| | Modality Shift OoD: X-Ray | | | | | | Question Type Shift OoD: Size | | | | | |
| | Closed-Ended | | | Open-Ended | | | Closed-Ended | | | Open-Ended | | |
| | i.i.d. | OoD | RR | i.i.d. | OoD | RR | i.i.d. | OoD | RR | i.i.d. | OoD | RR |
|---|---|---|---|---|---|---|---|---|---|---|---|---|
| No Finetuning | 0.48 | 0.35 | 0.73 | 1.73 | 1.85 | 1.07 | 0.46 | 0.59 | 1.28 | 1.77 | 2.03 | 1.15 |
| Prompt Tuning | 0.55 +/- 0.01 | 0.5 +/- 0.01 | 0.91 +/- 0.03 | 3.24 +/- 0.01 | 1.96 +/- 0.05 | 0.6 +/- 0.01 | 0.59 +/- 0.05 | 0.55 +/- 0.07 | 0.94 +/- 0.18 | 3.18 +/- 0.03 | 2.26 +/- 0.6 | 0.71 +/- 0.19 |
| LoRA | 0.64 +/- 0.03 | 0.47 +/- 0.07 | 0.73 +/- 0.11 | 3.25 +/- 0.03 | 2.03 +/- 0.04 | 0.62 +/- 0.02 | 0.69 +/- 0.01 | 0.53 +/- 0.18 | 0.76 +/- 0.25 | 3.13 +/- 0.02 | 2.95 +/- 0.0 | 0.94 +/- 0.01 |
| IA3 | 0.55 +/- 0.01 | 0.47 +/- 0.02 | 0.85 +/- 0.03 | 3.27 +/- 0.02 | 1.94 +/- 0.03 | 0.6 +/- 0.01 | 0.55 +/- 0.08 | 0.55 +/- 0.09 | 0.99 +/- 0.09 | 3.16 +/- 0.02 | 2.64 +/- 0.53 | 0.84 +/- 0.17 |

(b) Results for the non-medical base model.

| | Modality Shift OoD: X-Ray | | | | | | Question Type Shift OoD: Size | | | | | |
| | Closed-Ended | | | Open-Ended | | | Closed-Ended | | | Open-Ended | | |
| | i.i.d. | OoD | RR | i.i.d. | OoD | RR | i.i.d. | OoD | RR | i.i.d. | OoD | RR |
|---|---|---|---|---|---|---|---|---|---|---|---|---|
| No Finetuning | 0.43 | 0.32 | 0.75 | 1.5 | 1.61 | 1.08 | 0.4 | 0.35 | 0.88 | 1.53 | 2.03 | 1.32 |
| Prompt Tuning | 0.64 +/- 0.03 | 0.65 +/- 0.03 | 1.01 +/- 0.09 | 3.27 +/- 0.01 | 2.04 +/- 0.02 | 0.62 +/- 0.01 | 0.65 +/- 0.02 | 0.61 +/- 0.07 | 0.94 +/- 0.12 | 3.17 +/- 0.03 | 2.95 +/- 0.0 | 0.93 +/- 0.01 |
| LoRA | 0.67 +/- 0.01 | 0.46 +/- 0.06 | 0.68 +/- 0.08 | 3.25 +/- 0.02 | 2.01 +/- 0.04 | 0.62 +/- 0.01 | 0.68 +/- 0.01 | 0.69 +/- 0.03 | 1.01 +/- 0.06 | 3.15 +/- 0.02 | 2.26 +/- 0.6 | 0.72 +/- 0.19 |
| IA3 | 0.67 +/- 0.0 | 0.65 +/- 0.07 | 0.97 +/- 0.1 | 3.3 +/- 0.04 | 2.02 +/- 0.04 | 0.61 +/- 0.01 | 0.68 +/- 0.02 | 0.57 +/- 0.09 | 0.83 +/- 0.14 | 3.17 +/- 0.01 | 2.33 +/- 0.53 | 0.74 +/- 0.17 |

Table 16: **Robustness Results on the OVQA Dataset.** Results with ± indicate the mean and standard deviation over three seeds. Note that the most frequent baseline can only be calculated for the i.i.d. set as for OoD too few questions match the training set. RR: Relative Robustness.

(a) Results for the medical base model.

| | Body Part Shift OoD: Leg | | | | | | Question Type Shift OoD: Organ System | | | | | |
| | Closed-Ended | | | Open-Ended | | | Closed-Ended | | | Open-Ended | | |
| | i.i.d. | OoD | RR | i.i.d. | OoD | RR | i.i.d. | OoD | RR | i.i.d. | OoD | RR |
|---|---|---|---|---|---|---|---|---|---|---|---|---|
| No Finetuning | 0.4 | 0.4 | 0.99 | 2.27 | 2.27 | 1 | 0.4 | 0.38 | 0.93 | 2.11 | 2.32 | 1.1 |
| Full Finetuning | 0.72 | 0.58 | 0.8 | 3.16 | 2.19 | 0.69 | 0.76 | 0.08 | 0.1 | 2.98 | 1.25 | 0.42 |
| Prompt Tuning | 0.86 +/- 0.0 | 0.75 +/- 0.01 | 0.87 +/- 0.01 | 3.18 +/- 0.02 | 2.44 +/- 0.02 | 0.77 +/- 0.01 | 0.82 +/- 0.04 | 0.86 +/- 0.01 | 1.05 +/- 0.06 | 2.93 +/- 0.02 | 1.98 +/- 0.1 | 0.67 +/- 0.04 |
| LoRA | 0.86 +/- 0.01 | 0.77 +/- 0.0 | 0.9 +/- 0.01 | 3.27 +/- 0.01 | 2.49 +/- 0.02 | 0.76 +/- 0.01 | 0.84 +/- 0.02 | 0.74 +/- 0.06 | 0.88 +/- 0.07 | 3.09 +/- 0.04 | 1.98 +/- 0.15 | 0.64 +/- 0.06 |
| IA3 | 0.84 +/- 0.01 | 0.75 +/- 0.01 | 0.9 +/- 0.02 | 3.26 +/- 0.03 | 2.48 +/- 0.04 | 0.76 +/- 0.01 | 0.8 +/- 0.02 | 0.72 +/- 0.11 | 0.9 +/- 0.15 | 3.0 +/- 0.04 | 1.77 +/- 0.04 | 0.59 +/- 0.03 |
| Most Frequent | 0.75 | - | - | 2.62 | - | - | 0.73 | - | - | 2.27 | - | - |

(b) Results for the non-medical base model.

| | Body Part Shift OoD: Leg | | | | | | Question Type Shift OoD: Organ System | | | | | |
| | Closed-Ended | | | Open-Ended | | | Closed-Ended | | | Open-Ended | | |
| | i.i.d. | OoD | RR | i.i.d. | OoD | RR | i.i.d. | OoD | RR | i.i.d. | OoD | RR |
|---|---|---|---|---|---|---|---|---|---|---|---|---|
| No Finetuning | 0.42 | 0.46 | 1.09 | 1.8 | 2.05 | 1.14 | 0.4 | 0.7 | 1.74 | 1.85 | 1.9 | 1.03 |
| Full Finetuning | 0.82 | 0.74 | 0.91 | 3.21 | 2.3 | 0.72 | 0.75 | 0.08 | 0.1 | 2.94 | 1.47 | 0.5 |
| Prompt Tuning | 0.82 +/- 0.02 | 0.72 +/- 0.05 | 0.87 +/- 0.04 | 3.14 +/- 0.03 | 2.3 +/- 0.03 | 0.73 +/- 0.0 | 0.73 +/- 0.05 | 0.5 +/- 0.22 | 0.69 +/- 0.31 | 2.86 +/- 0.02 | 1.67 +/- 0.17 | 0.58 +/- 0.06 |
| LoRA | 0.82 +/- 0.02 | 0.74 +/- 0.01 | 0.9 +/- 0.01 | 3.2 +/- 0.08 | 2.4 +/- 0.01 | 0.75 +/- 0.02 | 0.8 +/- 0.02 | 0.31 +/- 0.04 | 0.39 +/- 0.05 | 3.05 +/- 0.09 | 1.31 +/- 0.09 | 0.43 +/- 0.02 |
| IA3 | 0.83 +/- 0.01 | 0.75 +/- 0.01 | 0.9 +/- 0.0 | 3.19 +/- 0.03 | 2.45 +/- 0.06 | 0.77 +/- 0.01 | 0.81 +/- 0.01 | 0.75 +/- 0.03 | 0.93 +/- 0.04 | 3.0 +/- 0.04 | 1.69 +/- 0.08 | 0.56 +/- 0.02 |
| Most Frequent | 0.75 | - | - | 2.62 | - | - | 0.73 | - | - | 2.27 | - | - |

Table 17: **No Image Baseline on the OVQA Dataset.** Results with ± indicate the mean and standard deviation over three seeds. The model was trained with the same methods as Table 16 just without seeing the image content. RR: Relative Robustness.

(a) Results for the medical base model.

| | Body Part Shift OoD: Leg | | | | | | Question Type Shift OoD: Organ System | | | | | |
| | Closed-Ended | | | Open-Ended | | | Closed-Ended | | | Open-Ended | | |
| | i.i.d. | OoD | RR | i.i.d. | OoD | RR | i.i.d. | OoD | RR | i.i.d. | OoD | RR |
|---|---|---|---|---|---|---|---|---|---|---|---|---|
| No Finetuning | 0.41 | 0.36 | 0.88 | 1.24 | 1.3 | 1.05 | 0.39 | 0.4 | 1.03 | 1.27 | 1.24 | 0.98 |
| Prompt Tuning | 0.75 +/- 0.01 | 0.7 +/- 0.01 | 0.94 +/- 0.01 | 2.66 +/- 0.1 | 2.09 +/- 0.02 | 0.79 +/- 0.02 | 0.67 +/- 0.03 | 0.44 +/- 0.01 | 0.66 +/- 0.02 | 2.32 +/- 0.05 | 1.39 +/- 0.07 | 0.6 +/- 0.04 |
| LoRA | 0.74 +/- 0.0 | 0.7 +/- 0.0 | 0.93 +/- 0.01 | 2.69 +/- 0.13 | 2.14 +/- 0.01 | 0.8 +/- 0.04 | 0.73 +/- 0.0 | 0.43 +/- 0.03 | 0.6 +/- 0.04 | 2.34 +/- 0.05 | 1.38 +/- 0.08 | 0.59 +/- 0.04 |
| IA3 | 0.74 +/- 0.0 | 0.69 +/- 0.02 | 0.93 +/- 0.02 | 2.75 +/- 0.05 | 2.12 +/- 0.01 | 0.77 +/- 0.02 | 0.7 +/- 0.04 | 0.39 +/- 0.06 | 0.56 +/- 0.1 | 2.32 +/- 0.03 | 1.33 +/- 0.07 | 0.57 +/- 0.03 |

(b) Results for the non-medical base model.

| | Body Part Shift OoD: Leg | | | | | | Question Type Shift OoD: Organ System | | | | | |
| | Closed-Ended | | | Open-Ended | | | Closed-Ended | | | Open-Ended | | |
| | i.i.d. | OoD | RR | i.i.d. | OoD | RR | i.i.d. | OoD | RR | i.i.d. | OoD | RR |
|---|---|---|---|---|---|---|---|---|---|---|---|---|
| No Finetuning | 0.2 | 0.19 | 0.95 | 1.21 | 1.15 | 0.95 | 0.19 | 0.35 | 1.83 | 1.14 | 1.33 | 1.16 |
| Prompt Tuning | 0.73 +/- 0.01 | 0.67 +/- 0.0 | 0.92 +/- 0.02 | 2.74 +/- 0.07 | 2.02 +/- 0.04 | 0.74 +/- 0.02 | 0.69 +/- 0.04 | 0.47 +/- 0.05 | 0.68 +/- 0.08 | 2.33 +/- 0.06 | 1.59 +/- 0.18 | 0.68 +/- 0.06 |
| LoRA | 0.73 +/- 0.03 | 0.68 +/- 0.0 | 0.93 +/- 0.02 | 2.69 +/- 0.14 | 2.12 +/- 0.01 | 0.79 +/- 0.04 | 0.73 +/- 0.0 | 0.43 +/- 0.01 | 0.6 +/- 0.02 | 2.32 +/- 0.03 | 1.35 +/- 0.04 | 0.58 +/- 0.03 |
| IA3 | 0.72 +/- 0.03 | 0.66 +/- 0.03 | 0.93 +/- 0.02 | 2.63 +/- 0.12 | 2.08 +/- 0.03 | 0.79 +/- 0.03 | 0.72 +/- 0.02 | 0.36 +/- 0.08 | 0.5 +/- 0.11 | 2.36 +/- 0.01 | 1.3 +/- 0.1 | 0.55 +/- 0.04 |

Table 18: **Robustness Results on the MIMIC Dataset.** Results with ± indicate the mean and standard deviation over three seeds. Note that the most frequent baseline can not be calculated as too few questions match the training set. RR: Relative Robustness.

(a) Results for the medical base model.

| | Gender Shift OoD: Female | | | | | | Ethnicity Shift OoD: Non-white | | | | | | Age Shift OoD: Young | | | | | |
| | Closed-Ended | | | Open-Ended | | | Closed-Ended | | | Open-Ended | | | Closed-Ended | | | Open-Ended | | |
| | i.i.d. | OoD | RR | i.i.d. | OoD | RR | i.i.d. | OoD | RR | i.i.d. | OoD | RR | i.i.d. | OoD | RR | i.i.d. | OoD | RR |
|---|---|---|---|---|---|---|---|---|---|---|---|---|---|---|---|---|---|---|
| No Finetuning | 0.51 | 0.5 | 0.97 | 1.81 | 1.77 | 0.97 | 0.51 | 0.5 | 0.99 | 1.86 | 1.77 | 0.95 | 0.52 | 0.48 | 0.93 | 1.86 | 1.8 | 0.97 |
| Full Finetuning | 0.75 | 0.75 | 1 | 3.41 | 3.43 | 1.01 | 0.73 | 0.78 | 1.07 | 3.26 | 3.54 | 1.09 | 0.71 | 0.79 | 1.1 | 3.27 | 3.51 | 1.07 |
| Prompt Tuning | 0.48 +/- 0.09 | 0.48 +/- 0.08 | 1.01 +/- 0.04 | 3.27 +/- 0.04 | 3.28 +/- 0.05 | 1.0 +/- 0.0 | 0.53 +/- 0.15 | 0.63 +/- 0.16 | 1.2 +/- 0.06 | 3.13 +/- 0.11 | 3.37 +/- 0.28 | 1.07 +/- 0.05 | 0.49 +/- 0.17 | 0.62 +/- 0.13 | 1.33 +/- 0.29 | 3.19 +/- 0.03 | 3.33 +/- 0.08 | 1.05 +/- 0.01 |
| LoRA | 0.75 +/- 0.0 | 0.76 +/- 0.0 | 1.02 +/- 0.01 | 3.38 +/- 0.07 | 3.41 +/- 0.04 | 1.01 +/- 0.01 | 0.71 +/- 0.03 | 0.79 +/- 0.02 | 1.11 +/- 0.02 | 3.31 +/- 0.03 | 3.6 +/- 0.01 | 1.09 +/- 0.01 | 0.73 +/- 0.01 | 0.79 +/- 0.01 | 1.07 +/- 0.0 | 3.34 +/- 0.03 | 3.58 +/- 0.06 | 1.07 +/- 0.01 |
| IA3 | 0.61 +/- 0.15 | 0.63 +/- 0.12 | 1.04 +/- 0.06 | 3.33 +/- 0.02 | 3.36 +/- 0.04 | 1.01 +/- 0.01 | 0.61 +/- 0.05 | 0.7 +/- 0.05 | 1.15 +/- 0.02 | 3.25 +/- 0.03 | 3.52 +/- 0.03 | 1.08 +/- 0.02 | 0.58 +/- 0.1 | 0.74 +/- 0.06 | 1.3 +/- 0.15 | 3.2 +/- 0.09 | 3.41 +/- 0.2 | 1.06 +/- 0.03 |

(b) Results for the non-medical base model.

| | Gender Shift OoD: Female | | | | | | Ethnicity Shift OoD: Non-white | | | | | | Age Shift OoD: Young | | | | | |
| | Closed-Ended | | | Open-Ended | | | Closed-Ended | | | Open-Ended | | | Closed-Ended | | | Open-Ended | | |
| | i.i.d. | OoD | RR | i.i.d. | OoD | RR | i.i.d. | OoD | RR | i.i.d. | OoD | RR | i.i.d. | OoD | RR | i.i.d. | OoD | RR |
|---|---|---|---|---|---|---|---|---|---|---|---|---|---|---|---|---|---|---|
| No Finetuning | 0.53 | 0.54 | 1.03 | 1.58 | 1.6 | 1.01 | 0.51 | 0.55 | 1.07 | 1.62 | 1.67 | 1.03 | 0.52 | 0.55 | 1.07 | 1.61 | 1.68 | 1.05 |
| Full Finetuning | 0.59 | 0.61 | 1.02 | 2.6 | 2.68 | 1.03 | 0.5 | 0.58 | 1.09 | 2.38 | 2.6 | 1.09 | 0.53 | 0.33 | 0.62 | 2.62 | 2.37 | 0.91 |
| Prompt Tuning | 0.66 +/- 0.02 | 0.66 +/- 0.03 | 1.01 +/- 0.02 | 3.14 +/- 0.06 | 3.15 +/- 0.04 | 1.0 +/- 0.01 | 0.55 +/- 0.05 | 0.63 +/- 0.02 | 1.16 +/- 0.1 | 2.98 +/- 0.05 | 3.18 +/- 0.23 | 1.07 +/- 0.06 | 0.63 +/- 0.02 | 0.51 +/- 0.15 | 0.8 +/- 0.21 | 3.1 +/- 0.01 | 3.05 +/- 0.09 | 0.99 +/- 0.03 |
| LoRA | 0.74 +/- 0.01 | 0.75 +/- 0.01 | 1.02 +/- 0.01 | 3.43 +/- 0.0 | 3.47 +/- 0.03 | 1.01 +/- 0.01 | 0.69 +/- 0.02 | 0.77 +/- 0.01 | 1.12 +/- 0.02 | 3.37 +/- 0.02 | 3.68 +/- 0.02 | 1.09 +/- 0.01 | 0.69 +/- 0.06 | 0.77 +/- 0.05 | 1.12 +/- 0.03 | 3.37 +/- 0.03 | 3.6 +/- 0.08 | 1.07 +/- 0.02 |
| IA3 | 0.69 +/- 0.06 | 0.72 +/- 0.05 | 1.04 +/- 0.02 | 3.31 +/- 0.02 | 3.35 +/- 0.01 | 1.01 +/- 0.01 | 0.68 +/- 0.03 | 0.78 +/- 0.01 | 1.14 +/- 0.03 | 3.21 +/- 0.02 | 3.51 +/- 0.03 | 1.09 +/- 0.0 | 0.65 +/- 0.09 | 0.71 +/- 0.12 | 1.1 +/- 0.04 | 3.15 +/- 0.16 | 3.33 +/- 0.08 | 1.06 +/- 0.05 |

Table 19: **No Image Baseline on the MIMIC Dataset.** Results with ± indicate the mean and standard deviation over three seeds. The model was trained with the same methods as Table 18 just without seeing the image content. RR: Relative Robustness.

(a) Results for the medical base model.

| | Gender Shift OoD: Female | | | | | | Ethnicity Shift OoD: Non-white | | | | | | Age Shift OoD: Young | | | | | |
| | Closed-Ended | | | Open-Ended | | | Closed-Ended | | | Open-Ended | | | Closed-Ended | | | Open-Ended | | |
| | i.i.d. | OoD | RR | i.i.d. | OoD | RR | i.i.d. | OoD | RR | i.i.d. | OoD | RR | i.i.d. | OoD | RR | i.i.d. | OoD | RR |
|---|---|---|---|---|---|---|---|---|---|---|---|---|---|---|---|---|---|---|
| No Finetuning | 0.51 | 0.49 | 0.96 | 1.5 | 1.48 | 0.99 | 0.51 | 0.5 | 0.98 | 1.56 | 1.49 | 0.96 | 0.53 | 0.47 | 0.89 | 1.58 | 1.48 | 0.94 |
| Prompt Tuning | 0.54 +/- 0.0 | 0.52 +/- 0.0 | 0.95 +/- 0.0 | 2.74 +/- 0.01 | 2.63 +/- 0.01 | 0.96 +/- 0.0 | 0.55 +/- 0.03 | 0.42 +/- 0.05 | 0.77 +/- 0.08 | 2.73 +/- 0.03 | 2.46 +/- 0.01 | 0.9 +/- 0.01 | 0.6 +/- 0.0 | 0.34 +/- 0.01 | 0.57 +/- 0.01 | 2.9 +/- 0.05 | 2.25 +/- 0.01 | 0.79 +/- 0.01 |
| LoRA | 0.55 +/- 0.0 | 0.51 +/- 0.0 | 0.94 +/- 0.0 | 2.75 +/- 0.01 | 2.64 +/- 0.01 | 0.96 +/- 0.0 | 0.55 +/- 0.0 | 0.42 +/- 0.0 | 0.76 +/- 0.0 | 2.74 +/- 0.03 | 2.45 +/- 0.02 | 0.89 +/- 0.01 | 0.59 +/- 0.01 | 0.35 +/- 0.02 | 0.6 +/- 0.04 | 2.9 +/- 0.05 | 2.25 +/- 0.01 | 0.78 +/- 0.01 |
| IA3 | 0.55 +/- 0.01 | 0.52 +/- 0.01 | 0.95 +/- 0.01 | 2.75 +/- 0.02 | 2.65 +/- 0.01 | 0.96 +/- 0.01 | 0.55 +/- 0.0 | 0.42 +/- 0.0 | 0.77 +/- 0.01 | 2.72 +/- 0.02 | 2.48 +/- 0.02 | 0.91 +/- 0.01 | 0.6 +/- 0.0 | 0.34 +/- 0.01 | 0.56 +/- 0.01 | 2.92 +/- 0.01 | 2.32 +/- 0.03 | 0.8 +/- 0.01 |

(b) Results for the non-medical base model.

| | Gender Shift OoD: Female | | | | | | Ethnicity Shift OoD: Non-white | | | | | | Age Shift OoD: Young | | | | | |
| | Closed-Ended | | | Open-Ended | | | Closed-Ended | | | Open-Ended | | | Closed-Ended | | | Open-Ended | | |
| | i.i.d. | OoD | RR | i.i.d. | OoD | RR | i.i.d. | OoD | RR | i.i.d. | OoD | RR | i.i.d. | OoD | RR | i.i.d. | OoD | RR |
|---|---|---|---|---|---|---|---|---|---|---|---|---|---|---|---|---|---|---|
| No Finetuning | 0.43 | 0.42 | 0.98 | 1.21 | 1.22 | 1.01 | 0.43 | 0.38 | 0.9 | 1.27 | 1.27 | 1 | 0.45 | 0.37 | 0.82 | 1.27 | 1.29 | 1.02 |
| Prompt Tuning | 0.54 +/- 0.0 | 0.51 +/- 0.0 | 0.95 +/- 0.0 | 2.72 +/- 0.01 | 2.64 +/- 0.01 | 0.97 +/- 0.0 | 0.53 +/- 0.03 | 0.45 +/- 0.05 | 0.85 +/- 0.14 | 2.72 +/- 0.04 | 2.46 +/- 0.04 | 0.9 +/- 0.02 | 0.5 +/- 0.08 | 0.49 +/- 0.12 | 1.03 +/- 0.44 | 2.9 +/- 0.02 | 2.32 +/- 0.02 | 0.8 +/- 0.0 |
| LoRA | 0.53 +/- 0.01 | 0.51 +/- 0.01 | 0.96 +/- 0.03 | 2.7 +/- 0.01 | 2.63 +/- 0.02 | 0.97 +/- 0.01 | 0.54 +/- 0.0 | 0.41 +/- 0.0 | 0.76 +/- 0.01 | 2.72 +/- 0.02 | 2.5 +/- 0.0 | 0.92 +/- 0.01 | 0.6 +/- 0.0 | 0.34 +/- 0.0 | 0.57 +/- 0.01 | 2.85 +/- 0.02 | 2.29 +/- 0.06 | 0.8 +/- 0.01 |
| IA3 | 0.46 +/- 0.11 | 0.46 +/- 0.12 | 1.01 +/- 0.02 | 2.72 +/- 0.02 | 2.67 +/- 0.02 | 0.98 +/- 0.01 | 0.54 +/- 0.0 | 0.45 +/- 0.04 | 0.84 +/- 0.07 | 2.71 +/- 0.0 | 2.5 +/- 0.0 | 0.92 +/- 0.0 | 0.59 +/- 0.01 | 0.34 +/- 0.02 | 0.58 +/- 0.01 | 2.87 +/- 0.01 | 2.32 +/- 0.02 | 0.81 +/- 0.01 |

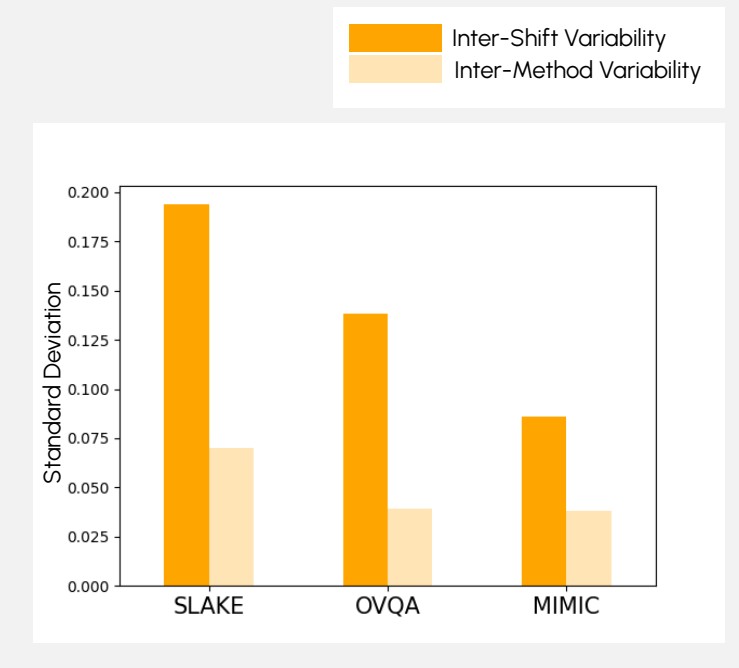

Figure 18: Standard deviation between shifts vs. standard deviation between PEFT methods for the medical base model, not including full FT. The type of shift has a higher impact on the robustness than the PEFT method.

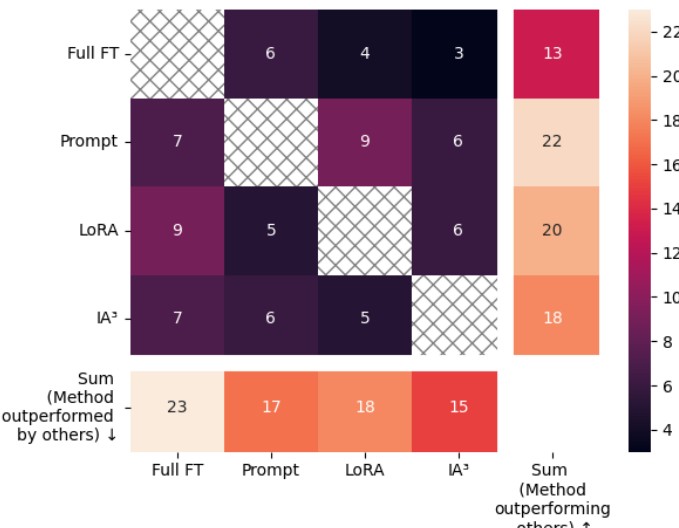

Figure 19: **Win/Loss Matrix for a pairwise comparison between FT methods.** The entries show cases where one FT method outperforms the other significantly, measures by a Welch's t-test with $p = 0.05$. A good method would achieve high values in its row (often outperforming others) and low values in its column (not often outperformed by others).

### C.5.1 Statistical analysis

In this section, we confirm two of the main findings in our robustness study by a detailed statistical analysis.

**No FT method consistently outperforms others in terms of robustness.** To prove this in addition to the rank analysis in Figure 8b), we run a pairwise Welch's t-test between the FT methods across the different shifts and correct for multiple testing. Each test-set is bootstrapped such that we have 100 robustness measures per experiment. We then report the Win/Loss Matrix with a p value of $p = 0.05$ (Bahri et al. (2022)) to see which FT method significantly outperforms another FT method significantly on a shift. The results are shown in Figure 19. While there is a slight trend indicating that Full FT is less likely to outperform other methods and is more frequently outperformed by them, the remaining methods show very similar performance levels. Overall, this analysis suggests that there is no clear winner of a method that more often outperforms others and is at the same time not often outperformed.

**Robustness trends are more stable across FT methods than across distribution shifts.** For this comparison, we conduct a one-way ANOVA on the robustness scores to compare the variability across FT methods versus across distribution shift types. The corresponding F-values and p-values are shown in Table 20. Confirming Figure 8c), we see that generally, the variance between the shifts is significantly higher than between methods ($F > 1, p < 0.05$). The only outlier is OVQA with a one shift being instable for Full FT, but this trend changes when excluding Full FT as also seen in Figure 18.

Table 20: One way ANOVA to analyze the variance between shifts and variance between methods.

| Dataset | All FT methods | | w\o Full FT | |
|---|---|---|---|---|
| | **F-value** | **p-value** | **F-Value** | **p-value** |
| SLAKE | 19.16 | $7.17 \times 10^{-5}$ | 16.83 | 0.0008 |
| OVQA | 1.11 | 0.38 | 21.71 | 0.0003 |
| MIMIC | 5.92 | 0.002 | 5.84 | 0.006 |

# D   Corruption Study

This section discusses the details of the corruption study showing the empirical confirmation of R1 (see section 3.3.1). Figure 20 shows the visualization of the corruption study pipeline, illustrating how artificial and realistic OoD test sets are generated and used to evaluate model robustness. We use OpenCV for image corruptions with the settings shown in Table 21.

Table 21: Corruption settings for the artificial shifts. Brackets indicate the altered parameter for each corruption, [...,...] indicate ranges for the corruption where randomly a value in that range is chosen.

|        | Blur (Kernel Size) | Gaussian Noise (Mean) | Brightness (Alpha) |
|--------|--------------------|------------------------|--------------------|
| Low    | 5                  | [0, 0.06]              | [1.1, 2]           |
| Medium | 7                  | [0.09, 0.15]           | [2.5, 4]           |
| High   | 11                 | [0.18, 0.25]           | [4.5, 6]           |

Table 22 shows the relative robustness results for both artificial and realistic shifts. The results show that both modality shift and question type shift exhibit lower relative robustness compared to all artificial shifts at low, medium, and high strengths. This suggests that artificial shifts, such as image corruption, fail to accurately represent the challenges posed by real-world, realistic shifts. The most prominent example here is the relative robustness of closed-ended questions under the artificial corruption shift which is up to 96% compared to the realistic shift (question type) which only has 61%. The only exception where the realistic shift shows higher robustness is the question type shift on the open-ended questions.

Table 22: Robustness results for the artificial and realistic shifts on SLAKE dataset

| | Corruption Shift (OoD: Corrupted i.i.d images) | | | | | | Corruption Shift (OoD: Corrupted i.i.d images) | | | | | |
| | Closed Ended | | | Open Ended | | | Closed Ended | | | Open Ended | | |
| | i.i.d. | OoD | RR | i.i.d. | OoD | RR | i.i.d. | OoD | RR | i.i.d. | OoD | RR |
|---|---|---|---|---|---|---|---|---|---|---|---|---|
| **Low Corruption** | $0.85 \pm 0.01$ | $0.83 \pm 0.01$ | $0.98 \pm 0.0$ | $4.36 \pm 0.03$ | $4.25 \pm 0.04$ | $0.98 \pm 0.02$ | $0.87 \pm 0.02$ | $0.84 \pm 0.0$ | $0.96 \pm 0.02$ | $4.27 \pm 0.01$ | $4.2 \pm 0.03$ | $0.98 \pm 0.01$ |
| **Medium Corruption** | $0.85 \pm 0.01$ | $0.79 \pm 0.02$ | $0.94 \pm 0.02$ | $4.36 \pm 0.03$ | $4.06 \pm 0.07$ | $0.93 \pm 0.02$ | $0.87 \pm 0.02$ | $0.82 \pm 0.01$ | $0.94 \pm 0.01$ | $4.27 \pm 0.01$ | $4.02 \pm 0.04$ | $0.94 \pm 0.01$ |
| **High Corruption** | $0.85 \pm 0.01$ | $0.74 \pm 0.01$ | $0.87 \pm 0.01$ | $4.36 \pm 0.03$ | $3.84 \pm 0.08$ | $0.88 \pm 0.03$ | $0.87 \pm 0.02$ | $0.76 \pm 0.02$ | $0.87 \pm 0.03$ | $4.27 \pm 0.01$ | $3.94 \pm 0.03$ | $0.92 \pm 0.01$ |

| | Modality shift (OoD: X-Ray) | | | | | | Question Type Shift (OoD: Size) | | | | | |
| | Closed Ended | | | Open Ended | | | Closed Ended | | | Open Ended | | |
| | i.i.d. | OoD | RR | i.i.d. | OoD | RR | i.i.d. | OoD | RR | i.i.d. | OoD | RR |
|---|---|---|---|---|---|---|---|---|---|---|---|---|
| **Realistic Shift** | $0.85 \pm 0.01$ | $0.64 \pm 0.07$ | $0.75 \pm 0.08$ | $4.36 \pm 0.03$ | $3.46 \pm 0.12$ | $0.79 \pm 0.03$ | $0.87 \pm 0.02$ | $0.53 \pm 0.06$ | $0.61 \pm 0.06$ | $4.27 \pm 0.01$ | $4.15 \pm 0.22$ | $0.97 \pm 0.05$ |

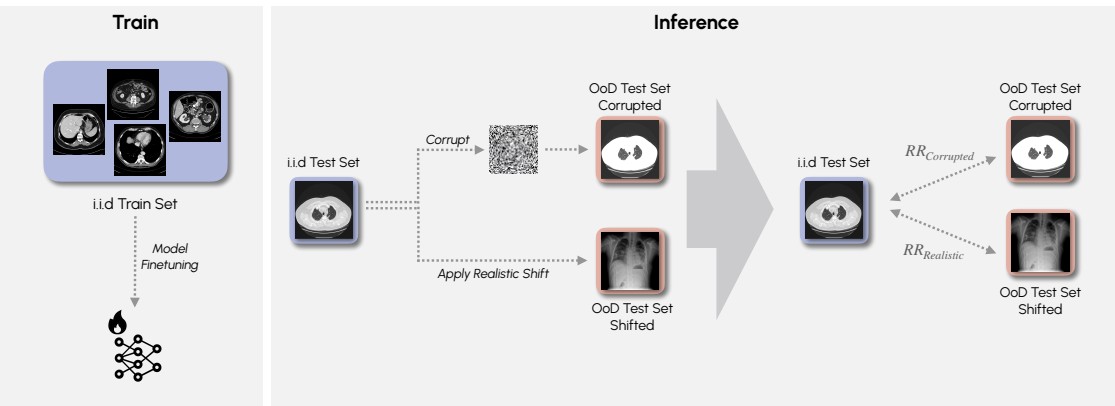

Figure 20: **Overview of the Corruption Study Pipeline.** The left side of the figure illustrates the training phase, where the model is fine-tuned on an i.i.d. training set. On the right, the inference phase depicts how two types of OoD test sets (artificial and realistic) are generated and used to evaluate robustness. For both the artificial and realistic shifts, the same i.i.d. test set is used. The corrupted OoD test set is generated by applying described artificial corruptions to the i.i.d. test images. The realistically shifted OoD test set is constructed using the data shifts introduced in section 3.1. Robustness is then measured by comparing the model's performance on the i.i.d. test set to its performance on each respective OoD set.

# E   Swapped Shifts

We conduct an ablation study on the SLAKE dataset to see how changing the i.i.d. and OoD set influences the robustness. Thereby, we use the same type of shifts as described in section 3.1 (Modality and Question Type), but with different i.i.d. and OoD sets. The concrete realization of the swapped shifts are shown in Table 23.

Table 23: Original and Swapped Shifts on the SLAKE Dataset.

| Shift Type | Original Shift | | Swapped Shift | |
|---|---|---|---|---|
| | i.i.d. | OoD | i.i.d. | OoD |
| Modality | CT/MRI | X-Ray | CT/X-Ray | MRI |
| Question Type | All questions except Size | Size | All questions except Position | Position |

The results are shown in Figure 21. Overall, there is a swap in closed-ended and open-ended robustness: While for the original shifts, the models showed less robustness on the closed-ended questions, in the swapped shifts, they show less robustness on the open-ended questions. As also noted in section 3.4, this arises from the related question patterns between certain OoD questions and the i.i.d. training data. Although the OoD questions are not explicitly seen during training, similar questions in the training set can provide enough contextual information for effective generalization. In this case, performance on the closed-ended OoD questions is higher because training examples like "Is this a study of the [body region]?" implicitly support OoD questions such as "Do the organs in the image exist in the [body region]?" This observation confirms one of our main findings: the type of shift has a greater impact on robustness than the choice of fine-tuning method. To achieve more reliable and generalizable performance, it is therefore more effective to focus on increasing the diversity of training data—so that it better reflects the target population—rather than solely optimizing the fine-tuning strategy.

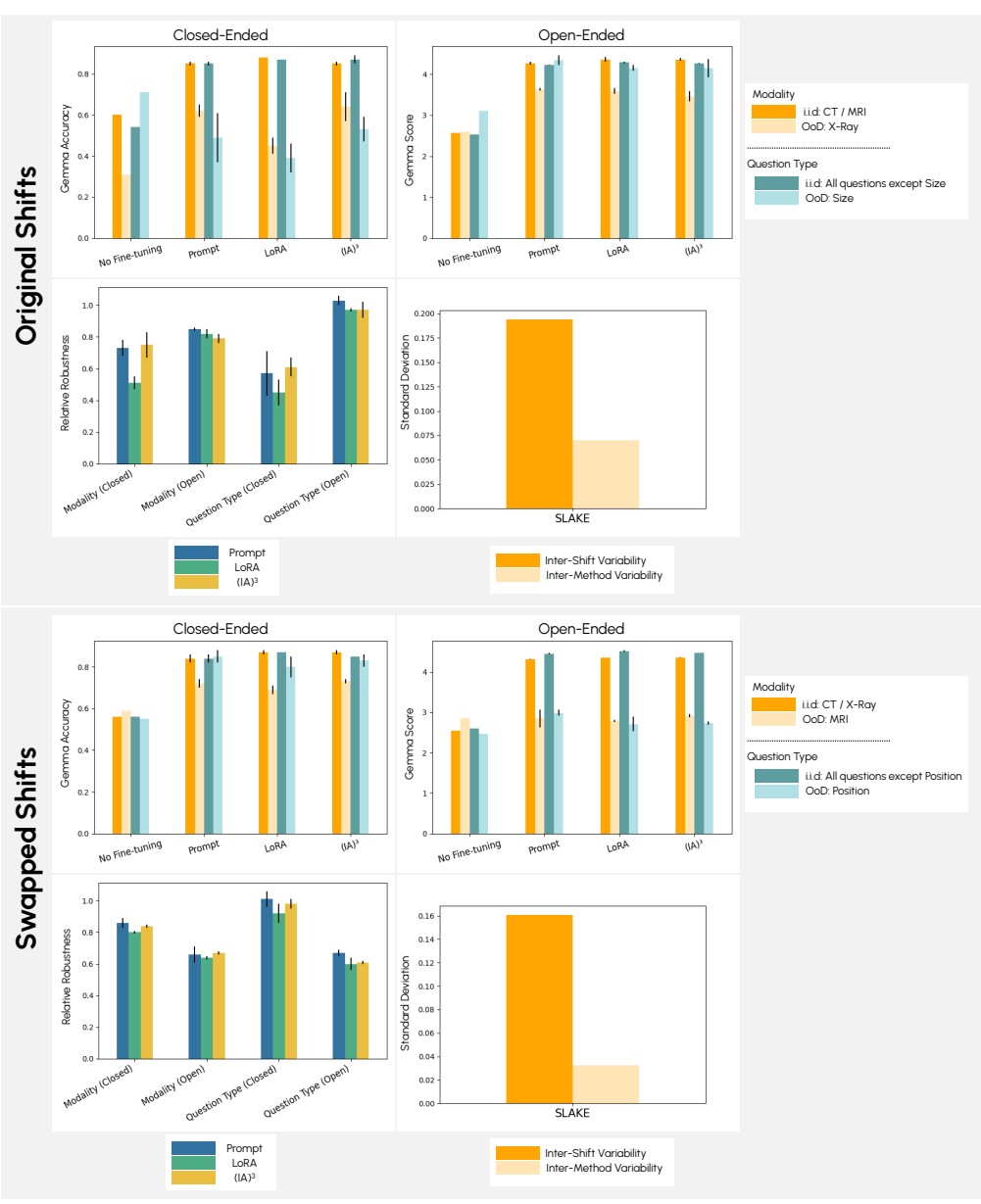

Figure 21: Results on the SLAKE dataset with swapped shifts, meaning the i.i.d. and OoD splits are changed. For each shift (Original Shifts, Swapped Shifts) the top row shows the i.i.d. and OoD performance on Open- and Closed-Ended questions, lower left shows the Relative Robustness, and lower right shows the Inter-Shift vs. Inter-Method variability.

## F  Multimodal Shifts

We conduct an ablation study on the OVQA dataset to evaluate the impact of a multimodal shift compared to the previously introduced unimodal shifts. Similar to the corruption study (Appendix D), this ablation study is performed on the medical base model with $(IA)^3$ as PEFT method. This multimodal shift combines the manifestation (Body Part) and question type shifts reported in our experiments. Specifically, we define the OoD set as samples featuring body part "Leg" and question type "Organ System", with all other samples classified as i.i.d. As shown in Figure 22, the multimodal shift demonstrates the lowest robustness compared to unimodal shifts, which is expected given that multimodal shifts represent a more extreme divergence than the unimodal ones.

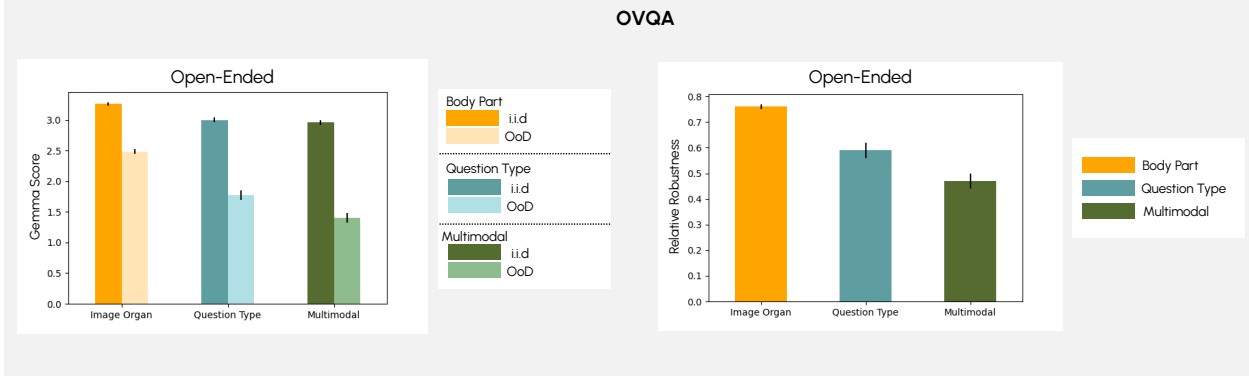

Figure 22: Performance results on OVQA dataset with image organ shift, question type shift, and multimodal shift which combines image organ shift and question type shift.

