# OpenReview forum: "SURE-VQA: Systematic Understanding of Robustness Evaluation in Medical VQA Tasks"
_TMLR — Accepted by TMLR_

### Review · Reviewer_SaK8 · 2025-05-08

**Summary Of Contributions:**

The paper mainly studies the robustness of Medical Visual Question Answering systems on realistic distribution shifts through the SURE-VQA framework. The framework introduces:

- Measuring Medical VQA performance on real world shifts (through modality shifting, body part shifting, question type shifting and population shifting) instead of synthetic corruption of adding noise, blur and brightness.
- The framework introduces a LLM based evaluation criteria for Medical VQA and shows its correlation with Human raters.
- The framework introduces basic sanity check baselines like no image baseline and most frequent baseline to determine how effectively the Medical VQA models use the input information or if they just use shortcuts in the dataset to get the answers.

The paper used this framework to observe the robustness of different fine-tuning methods across different shifts and datasets.

**Audience:**

Yes

**Broader Impact Concerns:**

The authors should mention that the framework should not be used as a measure of robustness to deploy Medical VQAs in real world situations yet.

**Claims And Evidence:**

Yes

**Requested Changes:**

- The authors should clearly define what they mean by open and close OoD.
- The authors should a reason of why they think the models perform well on Open ended question OoD.
- The authors say "Another study by Parcalabescu & Frank (2023) contextualizes the multimodal use of VLMs by employing Shapley values to assess the contribution of each modality to the output." Can the authors explain why this is not good enough to understand the contribution of images in answering the questions?
- In figure 3 the authors mention that they used iid with one of noise, blur or brightness corruption for medium corruption results. However for OoD shifts they use iid with one or more corruptions along with OoD samples. However, wouldn't using only OoD samples give us a better idea of the models performance on OoD samples? Also why is the corruption in iid samples increased to create OoDs? Doesn't this create similar synthetic OoDs in addition to the actual OoDs?
- The authors should provide an ablation on at least on one dataset where they change the training iids. For example in SLAKE train on CT and X-Ray and test on MRI. Or the authors should provide some explanation on what basis they chose the combination.
- The authors should clarify which fine-tuning method they are using for the initial experiments. (Fig 3 results)
- The authors should explicitly mention if they are testing the robustness of pre-trained VLMs or robustness of VLMs fine-tuned on certain datasets. For example, the most frequent baseline performing well is also dataset issue which leads to the model not getting trained to use the images effectively.

**Strengths And Weaknesses:**

Strengths:

- The paper provides thorough details on all the experiments and clearly explains the motivation behind them.
- The SURE-VQA method tackles the problem of determining robustness of a medical VQA system which is very important before real world deployment.

Weaknesses:
- The paper does not clearly explain the difference between open and close questions.
- The authors do not provide a reason for the consistent better performance on open oods.
- The authors do not explain how they decide on iids for training. For example, why use CT and MRI for training SLAKE and X-Ray for testing.
- The authors do not explicitly mention if they are testing the robustness for certain VLM architecture of if they are testing the robustness of VLMs fine-tuned on a certain dataset. In the second case the robustness also depends on the dataset and calls for creating better datasets (the most frequent baseline performing well is a problem of dataset for example). If the robustness of pretrained VLMs is measured here then that requires better training methods and pretraining data.

---

> ### Author Response · Authors · 2025-05-20
>
> Thank you again for your valuable comments, and for reading our general reply, as well as considering our point-by-point comments here:
>
> ---
> [...] should clearly define what they mean by open and close OoD.
> * Closed-ended questions are questions where a fixed set of options is given in the question (e.g. "CT"/"MRI"), while open-ended questions are not constrained to certain options.
> * We added this explanation to the beginning of section 3.1. and adapted the captions of Figure 5&6.
> ---
> [...] reason of why they think the models perform well on Open ended question OoD.
> * We believe this refers to our "Comparison Between Shifts" section, where we note high OoD performance on SLAKE open-ended questions under question type shift.
> * This might come from related patterns between the OoD test and i.i.d. training data. For instance, the SLAKE open-ended OoD question "What is the largest organ?" closely aligns with the training question "What is the main organ in the image?", whereas closed-ended OoD questions like "Which is smaller, [x] or [y]?" demand reasoning not covered during training.
> * This behavior is also discussed in an ablation study where we reversed the i.i.d. and OoD splits on SLAKE (Appendix E). The results support our finding: the type of shift impacts robustness more than the choice of FT method, so improving data diversity is more effective for generalization than only tuning FT strategies.
> * We clarified this in the manuscript part “Comparison Between Shifts”, Appendix E and "Future work”.
> ---
> [...] Another study [...] contextualizes the multimodal use of VLMs by employing Shapley values [...]  why this is not good enough [...]?
> * Thank you for your feedback. We do not argue that Shapley values are not good enough to understand the contribution of images in VQA. Rather, our approach is intended as a simpler, lightweight alternative.
> * One disadvantage of Shapley values is their high computational cost, as they require numerous forward passes. As reported in [1], Paragraph “A.2 Compute Requirements”, the analysis times ranged from 1.5-7 minutes per example. Given our setup of overall ~45.500 test samples, with 5 methods, 2 models, and 3 seeds, this would result in ~34.100 GPU hours, even with the lower estimate of 1.5 minutes per sample.
> * In summary, we chose our baselines for their simplicity and computational efficiency. Nonetheless, we agree that Shapley values offer an interesting perspective.
> * We added a discussion of this in Section 2 and mentioned it as future work in Section 4.
>
> [1] Parcalabescu et al. “Do Vision & Language Decoders use Images and Text equally? How Self-consistent are their Explanations?”, ICLR 2025
>
> ---
> In figure 3 the authors mention that they used iid with [...] corruption [...] However, wouldn't using only OoD samples give us a better idea of the models performance on OoD samples?[...]
> * We appreciate the feedback and believe there may be a misunderstanding regarding our corruption study. To clarify the setup, we added a new figure in Appendix D (Figure 20) and updated the corresponding text in Section 3.3.1 and the Figure 3 description.
> ---
> [...] should provide an ablation on at least on one dataset where they change the training iids. [...]
> * We agree that this provides an interesting additional perspective.Therefore, we added an ablation where we swap i.i.d. and OoD on the shifts of the SLAKE dataset to Appendix E of our paper. Please refer to our general reply for details.
> ---
> [...] clarify which fine-tuning method they are using for the initial experiments.
> * We adapted the caption of Figure 3 to clarify that we use (IA)^3, in addition to mentioning it in Section 3.3.1 “Study Design”.
> ---
> [...] explicitly mention if they are testing the robustness of pre-trained VLMs or robustness of VLMs fine-tuned [...] the most frequent baseline performing well is also dataset issue [...]
> * Thank you for your feedback. We mention that we test the robustness of fine-tuning methods in the abstract (“we conduct a study on the robustness of various FT methods”), introduction of section 3 (“conducting an extensive empirical study on the robustness of FT methods”), and the heading of section 3.4. However, if there is a section where this was unclear, we would appreciate it if you could indicate it, and we will revise it.
> * Regarding the dataset issue, we completely agree:  We see a strong need for more elaborate datasets to mitigate shortcut learning based on language, as discussed in Section 4, “Empirical Confirmation of R1–R3”. To emphasize it further, we added it also to “Future Work”.
> ---
> Broader Impact Concerns
> [...] should mention that the framework should not be used as a measure of robustness to deploy Medical VQAs in real world [...]
> * Thank you for your feedback. We have added a clarification to the Future Work section of the manuscript.
> ---
> We appreciate your feedback and believe we have resolved all issues. Please let us know in case you have any remaining concerns.

---

> > ### Comment · Reviewer_SaK8 · 2025-06-04
> > **No further comments**
> >
> > Thanks for the detailed response from the authors. I am okay with the response and have no further comments.

---

### Review · Reviewer_hWeX · 2025-05-10

**Summary Of Contributions:**

This paper propose Super-VQA, a medical VQA benchmark designed to systematically evaluate the robustness of vision-language models on medical VQA tasks. Specifically, Super-VQA is constructed with three key design principles: (1) leveraging naturally distributed data, (2) employing large language models for semantic-level answer evaluation, and (3) establishing robustness-focused baselines. Using this benchmark, the paper conduct a comprehensive robustness analysis across seven evaluation settings on three widely used medical VQA datasets (SLAKE, OVQA, and MIMIC-CXR-VQA) and four fine-tuning strategies (Full Fine-tuning, LoRA, Prompt Tuning, and (IA)³), yielding several key takeaways.

**Audience:**

Yes

**Claims And Evidence:**

Yes

**Requested Changes:**

**Q1**: Have the authors considered using more powerful LLMs, such as GPT-4o or Gemini 2.5 Pro, for the semantic-level evaluation?

**Strengths And Weaknesses:**

Strengths:
1. The paper addresses an important problem—how to mitigate common pitfalls in medical VQA evaluation—by proposing a robustness-oriented benchmark, which has practical value.
2. The experiments are comprehensive and large-scale. The resulting takeaways derived from the evaluations offer meaningful insights that can guide future research.
3. The authors provide open-source code, which facilitates reproducibility and adoption of the proposed Super-VQA benchmark.

Weakness:

1. The evaluation relies solely on Gemma for semantic-level assessment using LLMs. Due to the limitations of Gemma itself, the reliability of the evaluation results may be affected to some extent.

---

> ### Author Response · Authors · 2025-05-20
>
> Thank you again for your valuable comments, and for reading our general reply, as well as considering our point-by-point comments here:
>
> ---
>
> Q1: Have the authors considered using more powerful LLMs, such as GPT-4o or Gemini 2.5 Pro, for the semantic-level evaluation?
> - Thank you for the suggestion. While commercial closed-source models like GPT-4o or Gemini 2.5 Pro are frequently used, they come with several limitations: They are not open source, not freely accessible, and are subject to frequent updates, which can hinder reproducibility and long-term consistency in evaluation. For these reasons, we chose to use an open-source LLM as an evaluator. Importantly, results of our human rater study shows that Gemma already demonstrates high correlation with human judgments, highlighting its strong performance and suitability for our use case.
>
> ---
>
> We appreciate your feedback and believe we have resolved all issues. Please let us know in case you have any remaining concerns.

---

### Review · Reviewer_Euh2 · 2025-05-13

**Summary Of Contributions:**

The paper introduces SURE‑VQA, a three‑part evaluation framework designed to probe how well medical vision‑language models handle real world distribution shifts rather than the synthetic corruptions commonly used today.
1. Diverse, realistic splits (R1). The authors carve seven out‑of‑distribution scenarios—modality, body part, question type, sex, ethnicity, age, and combined shifts—across three public datasets.
2. Semantics‑aware scoring (R2). They replace surface‑form metrics like BLEU with a Gemma‑based rubric that assigns 1‑to‑5‑star ratings and falls back to exact match when answers coincide verbatim; a five‑rater study shows Gemma’s scores correlate best with human judgment.
3. Sanity baselines (R3). “Image‑blind” and “majority‑answer” baselines reveal how far a model can get on language priors alone, anchoring the robustness scores.

**Audience:**

Yes

**Claims And Evidence:**

Yes

**Requested Changes:**

See weakness

**Strengths And Weaknesses:**

### Strengths
- Targets real clinical pain‑points. The authors don’t settle for toy corruptions like random noise. Instead, they probe shifts clinicians actually face—different imaging devices, diverse patient populations, and varied question phrasings—so the lessons carry straight to the bedside.
- By mixing three datasets, seven shift categories, and four fine‑tuning strategies, the study chases patterns showing that no single tweak makes a model bullet‑proof.
-   All code, split scripts, and prompts are on GitHub, lowering the barrier for follow‑up work and making the paper a community touchstone.

### Weaknesses
-  Narrow imaging scope. Everything is radiology, so generality remains an open question
-  Single‑grader limitation. The study leans on Gemma alone to score answers; human spot‑checks help, but Gemma’s medical know‑how still has blind spots.
- Shallow sanity check. The “image‑blind” baseline tells us vision isn’t strictly required, yet there’s no Shapley‑style breakdown to show how much each modality (vision vs. text) really contributes.
- Lightweight statistics. Results come without confidence intervals or multiple‑comparison corrections, so it’s hard to judge which performance gaps are truly significant.

---

> ### Author Response · Authors · 2025-05-20
>
> Thank you again for your valuable comments, and for reading our general reply, as well as considering our point-by-point comments here:
>
> ---
>
> Narrow imaging scope. Everything is radiology, so generality remains an open question.
> - Thank you for bringing up this point. As also discussed in Section 4, Paragraph “Generalizability of the Framework”, we would like to see our work as a foundational framework, where adhering to the requirements proposed in our paper also generalizes to other image types.
> - However, we agree that extending the study to more medical image types is an interesting point of future work, as also discussed in section 4, Paragraph “Future Work”. We revised this part to also more clearly state that other medical image types might have different characteristics and thus may reveal other findings.
> ---
> Single‑grader limitation. The study leans on Gemma alone to score answers; human spot‑checks help, but Gemma’s medical know‑how still has blind spots.
> - Thank you for your feedback. We conducted a human-rater study comparing several open-source LLMs, including medical ones like BioMistral and BioMistral-DARE, and found that Gemma consistently aligned best with human judgments across datasets, outperforming both general-purpose and medical alternatives.
> - While all LLMs may carry blind spots from limited medical or demographic representation, we believe this is precisely why human-rater study is essential – to use human evaluation to detect and mitigate such blind spots.
> - That said, given Gemma’s strong agreement with expert ratings, we believe this validates its use as an evaluator in our study.
> ---
> Shallow sanity check. The “image‑blind” baseline tells us vision isn’t strictly required, yet there’s no Shapley‑style breakdown to show how much each modality (vision vs. text) really contributes.
> - Thank you for raising this point. Our goal was to propose a simple and computationally efficient way to probe modality reliance using our sanity baseline. Thereby, our approach is intended as a lightweight alternative to Shapley values.
> - One of the main limitations of Shapley-based methods is their high computational overhead [1], which requires numerous forward passes and can become impractical in large-scale empirical studies like ours. As reported in [1], Paragraph “A.2 Compute Requirements”, the analysis times ranged from 1.5-7 minutes per example (in extreme cases even 25 minutes for models operating on multiple resolutions). Given our setup of overall ~45.500 data samples in the test sets, with 5 methods, 2 models, and 3 seeds, this would result in a compute demand of ~34.100 GPU hours, even with the lower estimate of 1.5 minutes compute time per sample. In contrast, our baseline provides a clear and efficient signal of how much information the model can extract from language alone.
> - In summary, we chose our baselines for their simplicity and computational efficiency. Nonetheless, we agree that Shapley values offer an interesting perspective, and they could complement our approach in future work.
> - We have added a discussion of this point in Section 2 and mention it as a direction for future work in Section 4 (Future Work).
>
> [1] Parcalabescu and Frank. “Do Vision & Language Decoders use Images and Text equally? How Self-consistent are their Explanations?”, ICLR 2025
>
> ---
> Lightweight statistics. Results come without confidence intervals or multiple‑comparison corrections, so it’s hard to judge which performance gaps are truly significant.
>
> - Thank you for your feedback. In response, we have added a more rigorous statistical evaluation to support our main claims:
>     - **No FT method consistently outperforms others in robustness:**
>     To support this, we performed Welch's t-tests across FT methods on the robustness scores from bootstrapped samples. We present the results as a Win/Loss Matrix, using a significance threshold of p = 0.05 to identify whether one FT method significantly outperforms another on a given shift.  To account for multiple comparisons and improve the reliability of the findings, we applied the Benjamini/Hochberg (non-negative) correction.
>     - **Robustness trends are more stable across FT methods than across distribution shifts:**
>     For this comparison, we conducted a one-way ANOVA on the robustness scores to compare the variability across FT methods versus across distribution shift types.The corresponding F-values and p-values are reported to quantify the significance of these differences.
> - We have updated the main paper to reference these statistical findings where relevant and included full details of all tests and results in Appendix Section C.5.1.
> ---
>
> We appreciate your feedback and believe we have resolved all issues. Please let us know in case you have any remaining concerns.

---

### Author Response · Authors · 2025-05-20

We sincerely thank all reviewers for their valuable comments. The reviewers generally agreed on the added value of our work (“Targets real clinical pain‑points”, “The paper addresses an important problem”, “experiments are comprehensive and large-scale”, “thorough details on all the experiments and clearly explains the motivation behind them”, “provide open-source code, which facilitates reproducibility and adoption”).

Next to the point-by-point responses, we provide a summary of the main revisions made to the manuscript in response to the reviewers’ concerns here:

---

**Providing an additional ablation changing the i.i.d. and OoD shift** (raised by SaK8)
* We add an ablation where we swap the i.i.d. and OoD set on the shifts of the SLAKE dataset to Appendix E of our paper.
* While we see a change in the robustness of open- and closed-ended questions when changing the i.i.d and OoD sets, this ablation confirms our main insight that the shift has more influence on the robustness than the FT method used.
* The detailed results with an analysis are in Appendix E of our updated manuscript.

---

**Adding more statistical analysis to support main insights** (raised by Euh2)
* We have added a more rigorous statistical evaluation to support our main claims:
    * **No FT method consistently outperforms others in robustness:** To support this, we performed Welch's t-tests across FT methods on the robustness scores from bootstrapped samples. We present the results as a Win/Loss Matrix, using a significance threshold of p = 0.05 to identify whether one FT method significantly outperforms another on a given shift.  To account for multiple comparisons and improve the reliability of the findings, we applied the Benjamini/Hochberg (non-negative) correction.
    * **Robustness trends are more stable across FT methods than across distribution shifts:** For this comparison, we conducted a one-way ANOVA on the robustness scores to compare the variability across FT methods versus across distribution shift types.The corresponding F-values and p-values are reported to quantify the significance of these differences.
* We have updated the main paper to reference these statistical findings where relevant and included full details of all tests and results in Appendix Section C.5.1.

---

**General clarifications**
In order to address the remaining points, we have revised parts of our manuscript for clarifications. All revisions in the manuscript are highlighted in red. Changes include:
* Clearly defining the terminology of closed- and open-ended questions
* Qualitative analysis on the high OoD performance on certain shifts
* Discussing our sanity baselines in comparison to Shapley values
* Clarifying the corruption study setup
* Stating limitation that SURE-VQA is not meant for clinical deployment
* Emphasizing limitation about imaging scope

---

​​We believe these updates resolve the stated concerns. Please find our point-by-point answers in the respective reviewer sections.

---

### Decision · Action_Editor_39A8 · 2025-06-12

**Recommendation:** Accept as is

**Audience:**

Yes

**Audience Explanation:**

This paper will interest researchers developing medical vision-language models and those focused on evaluating robustness under real-world clinical shifts. It introduces a reproducible benchmark with semantic-level evaluation and highlights key challenges in generalization, fairness, and modality reliance.

**Claims And Evidence:**

Yes

**Claims Explanation:**

The paper presents SURE-VQA, a benchmark designed to evaluate the robustness of medical vision-language models (VLMs) under realistic distribution shifts. The work focuses on clinically meaningful shifts, such as imaging modality, patient demographics, and question types, and offers a semantic-level evaluation using an open-source LLM (Gemma), supported by a human study. The authors also introduce simple but effective baselines to assess modality reliance.

This submission is reviewed by three experts, who have provided valuable suggestions and insightful comments. All three reviewers acknowledge the importance of the problem and the clarity and scale of the empirical analysis. The authors’ responses during the rebuttal period were thorough, addressing all technical and conceptual concerns. The additional statistical analysis (Welch’s t-tests, ANOVA) and ablation on i.i.d./OoD splits helped strengthen the key claims.

Based on the above, I concur with the acceptance of three reviewers.